# A Simple, Cost-Effective, and Automation-Friendly Direct PCR Approach for Bacterial Community Analysis

Fangchao Song,[a] Jennifer V. Kuehl,[a] Arjun Chandran,[b] Adam P. Arkin[a,b]

[a]Environmental Genomics & Systems Biology Division, Lawrence Berkeley National Laboratory, Berkeley, California, USA
[b]Department of Bioengineering, University of California at Berkeley, Berkeley, California, USA

**ABSTRACT** Bacterial communities in water, soil, and humans play an essential role in environmental ecology and human health. PCR-based amplicon analysis, such as 16S rRNA sequencing, is a fundamental tool for quantifying and studying microbial composition, dynamics, and interactions. However, given the complexity of microbial communities, a substantial number of samples becomes necessary for analyses that parse the factors that determine microbial composition. A common bottleneck in performing these kinds of experiments is genomic DNA (gDNA) extraction, which is time-consuming, expensive, and often biased based on the types of species present. Direct PCR method is a potentially simpler and more accurate alternative to gDNA extraction methods that do not require the intervening purification step. In this study, we evaluated three variations of direct PCR methods using diverse heterogeneous bacterial cultures, including both Gram-positive and Gram-negative species, ZymoBIOMICS microbial community standards, and groundwater. By comparing direct PCR methods with DNeasy Blood and Tissue Kits for microbial isolates and DNeasy PowerSoil Kits for microbial communities, we found that a specific variant of the direct PCR method exhibits an overall efficiency comparable to that of the conventional DNeasy PowerSoil protocol in the circumstances we tested. We also found that the method showed higher efficiency for extracting gDNA from the Gram-negative strains compared to DNeasy Blood and Tissue protocol. This direct PCR method is 1,600 times less expensive ($0.34 for 96 samples) and 10 times simpler (15 min hands-on time for 96 samples) than the DNeasy PowerSoil protocol. The direct PCR method can also be fully automated and is compatible with small-volume samples, thereby permitting scaling of samples and replicates needed to support high-throughput large-scale bacterial community analysis.

**IMPORTANCE** Understanding bacterial interactions and assembly in complex microbial communities using 16S rRNA sequencing normally requires a large experimental load. However, the current DNA extraction methods, including cell disruption and genomic DNA purification, are normally biased, costly, time-consuming, labor-intensive, and not amenable to miniaturization by droplets or 1,536-well plates due to the significant DNA loss during the purification step for tiny-volume and low-cell-density samples. A direct PCR method could potentially solve these problems. In this study, we developed a direct PCR method which exhibits similar efficiency as the widely used method, the DNeasy PowerSoil protocol, while being 1,600 times less expensive and 10 times faster to execute. This simple, cost-effective, and automation-friendly direct-PCR-based 16S rRNA sequencing method allows us to study the dynamics, microbial interaction, and assembly of various microbial communities in a high-throughput fashion.

**KEYWORDS** 16S rRNA sequencing, microbial communities

Address correspondence to Fangchao Song, fsong@lbl.gov, or Adam P. Arkin, aparkin@lbl.gov.

The microbial communities that populate water, soil, and animals drive complex ecological processes and play influential roles in ecosystem services and health. Studying these processes often starts by assessing bacterial diversity and identifying

**TABLE 1** List of cell disruption methods, functions, and PCR compatibility

| Cell lysis methods[a] | Function | PCR compatibility | References |
|---|---|---|---|
| **Mechanical** | | | |
| Bead beating | Mechanically destroy cell membrane structure | Yes | 34, 35 |
| Freeze-thaw | Cell membrane cracking by ice crystals | Yes | 34, 36 |
| **Chemical** | | | |
| SDS, anionic surfactant | Disrupt the cell membrane phospholipids (strong) | Generally, no; conditionally, yes | 34, 37 |
| Sodium lauroyl sarcosinate (Sarkosyl), anionic surfactant | Disrupt the cell membrane phospholipids (strong) | No | 38, 39 |
| CTAB, quaternary ammonium surfactant | Disrupt the cell membrane phospholipids (strong) | Unknown | 34, 40 |
| IGEPAL CA-630, nonionic surfactant | Disrupt the cell membrane phospholipids (mild) | Yes | 19, 20 |
| Triton X-100, nonionic surfactant | Disrupt the cell membrane phospholipids (mild) | Yes | 34, 41 |
| CHAPS, zwitterionic surfactant | Disrupt cell membrane | No | 41, 42 |
| Potassium hydroxide, alkaline | Disrupt cell membrane (strong) | No | 43, 44 |
| **Enzymatic** | | | |
| Lysozyme | Disrupt peptidoglycan | No | 34, 45 |
| Lysostaphin | Disrupt peptidoglycan | Unknown | 46, 47 |
| Proteinase K | Disrupt membrane protein | Yes, by deactivation | 18, 34 |
| **Heat** | | | |
| Boiling | Disrupt cell membrane and protein denaturation | Yes | 48, 49 |

[a]CTAB, cetyl trimethylammonium bromide; CHAPS, 3-cholamidopropyl dimethylammonio-1-propanesulfonate.

bacterial species. For complex communities, this is most often accomplished by amplifying 16S rRNA from microbiome samples with PCR targeting specified variable regions, sequencing the complex mixture of molecules, and calculating the relative abundance of the inferred distinct ribosome genes (16S sequencing) (1–3). It is often the goal to use these data to find associations between community composition and biological function. However, this is hampered by a number of complexities in the analysis and interpretation of these data arising at multiple stages of the process. These can range from biased extraction and amplification of nucleic acids from different types of bacteria in different growth phases to problems of abundance estimation. Among them, DNA extraction is considered to lead to the most striking bias between previously tested protocols (4, 5). Moreover, research on microbiomes is rapidly expanding, while the cost of such research, due to the widespread use of laboratory automation and the progress of next-generation sequencing, is decreasing. Large-scale microbial community analysis is hampered by the time and labor costs of DNA extraction and purification (6). In addition, as high-throughput bacterial assays get smaller, such as in droplet or in well-plate-based assays, current column- or bead-based DNA purification method are ineffective due to significant DNA loss for low-cell-number samples (7–10). Therefore, it is imperative to design a new DNA extraction and amplification method which is efficient, cost-effective, and amenable to miniaturization.

There have been a number of DNA extraction approaches to increase the general efficiency of extracting nucleic acids (DNA and RNA) from all cells in a sample (11–14). These methods differ based on organism type, sample materials (e.g., sediment versus water), and compatibility with downstream processing. Ideally, cell disruption and DNA amplification could be done without an intervening purification step, and the entire set of operations should be amenable to miniaturization and automation. However, these direct PCR techniques, while popular, when applied to bacteria are thought to be nonquantitative, and worse than the nondirect PCR methods, due to the limited choices of bacterial cell disruption methods compatible with PCR chemistries. Of those that have become popular, very few have been rigorously tested for both efficiency and precision in the application of 16S sequencing of microbial communities (15, 16). None of the direct PCR methods have been quantified by comparing the obtained composition with the real composition (of a standard community) or optimized for automation or miniaturization. Table 1 lists some of the basal cell disruption

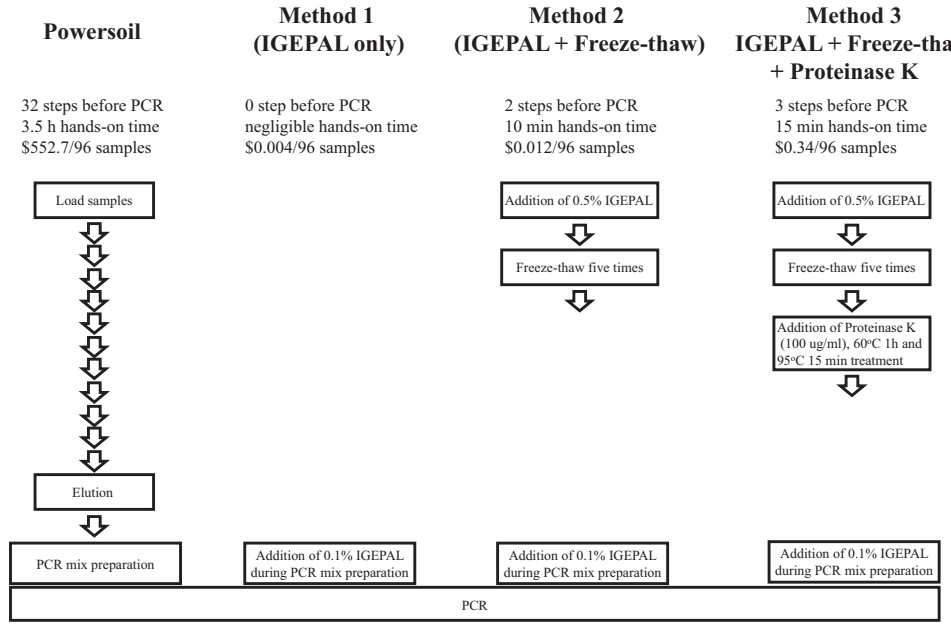

**FIG 1** Workflows of conventional DNA extraction method and direct PCR methods for analyzing microbial communities.

methods used for DNA extraction, along with their functions and PCR compatibility. Among them, alkaline and bead-beating are very effective universal disruptors, but DNA released early during these processes may be damaged by the duration of treatments necessary to extract DNA from more recalcitrant cells (17). Sodium dodecyl sulfate (SDS) is very effective for Gram-negative bacteria, but its use generally requires a purification step before PCR amplification of the sample (18). Only a small portion of these methods are compatible with direct PCR, of which very few have been directly tested on samples of known composition with sufficient diversity to uncover biases in extraction. Thus, we propose that the combination of a few of the basal DNA disruption methods would fulfill the needs of an effective direct PCR protocol for extraction and amplification of 16S RNA from bacterial microbiome samples.

We sought an effective combination of DNA extraction techniques that was inexpensive, easy, and compatible with direct PCR to reduce bias and increase scalability. To do so, we explored the use of a PCR-compatible nonionic surfactant (IGEPAL CA-630) that has been successfully applied in eukaryotic proteomics and RNA-SEQ studies (19, 20) with other PCR-compatible techniques, such as freeze-thaw cycles, proteinase K treatment, and variation in heating time. We compared the performance of IGEPAL treatments with different combinations of these three membrane disruption methods to each other, and two commercial kits, the DNeasy Blood and Tissue Kit and the DNeasy PowerSoil DNA Isolation Kit (PowerSoil), which are two of the most widely used methods for effective DNA extraction. To assess performance, we tested the methods on a mock community designed to encompass bacteria with different cellular properties in different phases of growth and on the more diverse groundwater communities. We found that the best combination of our approach yields overall quantification comparable to that of the PowerSoil method, though biases still persist (like with the PowerSoil method), but with a far shorter and more cost-effective protocol that is compatible with miniaturization and automation.

## RESULTS

We evaluated the three new protocols shown in Fig. 1. The IGEPAL-only method (method 1) uses the surfactant (IGEPAL CA-630) and longer heating than current protocols (10 min at 98°C for initial activation during PCR); the IGEPAL+freeze-thaw method

**TABLE 2** Comparison of DNA extraction and direct PCR methods

| Method | Cost/96 samples (USD)[a] | Protocol steps[b] | Extraction time[c] | Hands-on time |
|---|---|---|---|---|
| DNeasy PowerSoil HTP 96 kit | $552.70 | 32 | 4 h | 3.5 h |
| Extract-N-Amp plant PCR kit | $176.90 | 3 | 25 min | 15 min |
| Method 1 (IGEPAL only) | $0.004 | None | Negligible | Negligible |
| Method 2 (IGEPAL+freeze-thaw) | $0.012 | 2 | 45 min | 10 min |
| Method 3 (IGEPAL+freeze-thaw+proteinase K) | $0.34 | 3 | 2 h | 15 min |

[a]For calculation of the costs, see Table S3.
[b]"None" indicates that there is no extra step before addition of PCR reagents.
[c]Time is estimated for processing 96 samples, including waiting time.

(method 2) adds freeze-thaw sequence on top of method 1; the IGEPAL+freeze-thaw+proteinase K method (method 3) adds a proteinase K treatment and extra heating during proteinase K treatment on top of method 2. Since the three protocols use different combinations of mechanisms to disrupt membranes, we expected increasing extraction efficiency and decreasing bias as we went from method 1 to 3. As shown in Table 1, surfactant may disrupt the membrane phospholipids, proteinase K disrupts membrane proteins, and heating and freeze-thawing generally disrupt the bacterial membrane mechanically. To evaluate these methods, we first quantified the extraction efficiency on a set of specially chosen target bacteria with different membrane properties using quantitative PCR (qPCR). We then compared these methods to the PowerSoil protocol by quantifying the ability to operate in mixed culture and reproduce known abundance ratios using a specially designed mock community. We further compared the results from application to diverse groundwater communities sampled from the Bear Creek Valley watershed of Oak Ridge, TN. Finally, the costs and other aspects of these methods are presented in Table 2.

**Evaluation of direct PCR methods using model strains and qPCR.** The structures of microbial membranes and other elements that prevent DNA accessibility are extremely diverse among bacterial phyla and can even vary across the growth phases of a given species (21, 22). Therefore, it is almost inevitable that any method of DNA extraction will have a different efficacy across these factors. To understand the variation of efficacy, we started by testing our methods on four model strains—two Gram-negative strains (*Escherichia coli* and *Pseudomonas putida*) and two Gram positive strains (*Lactococcus lactis cremoris* MG1363 [referred to here as *Lactococcus lactis*] and *Lactobacillus brevis*)—in both exponential and stationary phases.

We first tested if IGEPAL alone could efficiently kill bacteria and be compatible with PCR. Stationary-phase cultures of our test bacteria (*Escherichia coli*, *Pseudomonas putida*, and *Lactococcus lactis*) were exposed to 0.1% IGEPAL at 98°C for 5 min and were plated on LB agar plates. No colonies appeared after 2 days' growth, implying full efficacy in killing the bacteria. We then tested PCR of three different 16S PCR primers using a standard protocol with either KAPA HotStart HiFi polymerase or *Taq* polymerase augmented with 0.1% IGEPAL. The gel image of PCR products shown in Fig. S1A indicates that the PCR was unperturbed by addition of the surfactant IGEPAL CA-630.

We then evaluated the ability of the three direct PCR methods to quantify the amount of genomic DNA (gDNA) in each exponential- and stationary-phase sample compared to one of the most popular commercial kits, the DNeasy Blood and Tissue Kit, which has been recommended by comparing with multiple commercially available kits, for complete gDNA extraction (11, 23). We compared the estimation of relative gDNA concentrations using the threshold cycle ($C_T$) of qPCR by the direct PCR methods versus the DNeasy Blood and Tissue Kit using the same cell cultures. Figure 2 shows the direct comparison of cycle threshold ($C_T$) values between gDNA extracted with the DNeasy Blood and Tissue Kit (gDNA control) and an equivalent amount of cells with the direct PCR methods. To better compare the difference between the gDNA control and the direct PCR methods, the

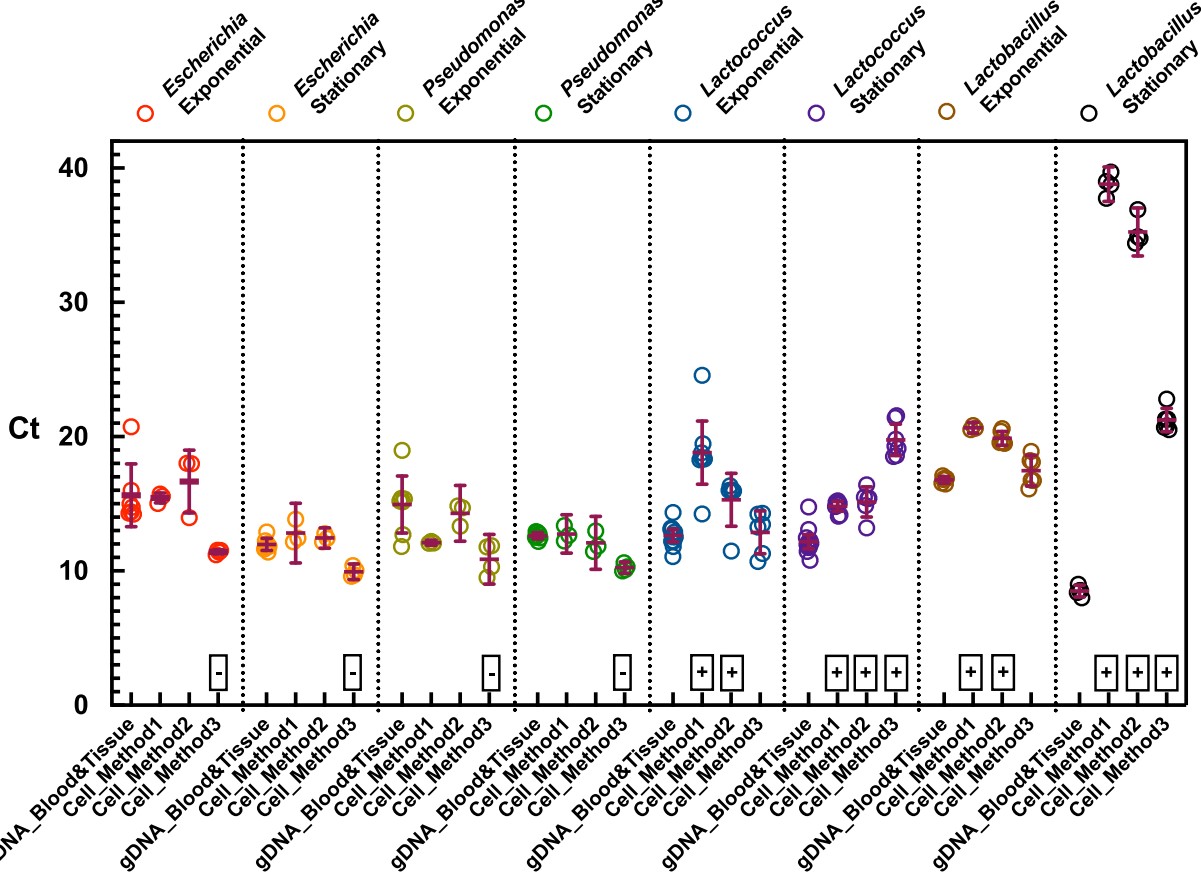

**FIG 2** $C_T$ from quantitative PCR (qPCR) of genomic DNA extracted by DNeasy Blood and Tissue Kits and cells treated by direct PCR methods (method 1, IGEPAL only; method 2, IGEPAL+freeze-thaw; method 3, IGEPAL+freeze-thaw+proteinase K). The strains are *Escherichia coli* K-12 MG1655, *Pseudomonas putida* KT2440, *Lactococcus lactis cremoris* MG1363, and *Lactobacillus brevis* ATCC 14869. A plus sign indicates that the $C_T$ of the direct PCR method is greater than the $C_T$ of the extracted gDNA; a minus sign indicates that the $C_T$ of the direct PCR method is less than the $C_T$ of the extracted gDNA; $C_T$ with no symbols are similar between extracted gDNA and direct PCR methods.

average difference of $C_T$ between direct PCR methods and gDNA control ($\Delta C_T = C_T$ of direct PCR $- C_T$ of gDNA control) and the $P$ value by $t$ test were calculated and are shown in Table S1. In Fig. 2, the cell types with a $\Delta C_T$ value of >0 and a $P$ value of <0.05 are labeled with plus signs, indicating that the direct PCR methods are less effective than the DNeasy Blood and Tissue Kit; the cell types with $\Delta C_T$ value of <0 and a $P$ value of <0.05 are labeled with minus signs, indicating that the direct PCR methods are more effective than the DNeasy Blood and Tissue Kit; the cell types with $P$ values of >0.05 are not labeled, which suggests that the direct PCR methods seem to be similarly effective to the DNeasy Blood and Tissue Kit.

Collectively, all three methods could effectively interrupt the Gram-negative bacteria (*Escherichia coli*, *Pseudomonas putida*) in both exponential and stationary phases, similar to or better than the DNeasy Blood and Tissue Kit. In particular, *Escherichia coli* and *Pseudomonas putida* in both exponential and stationary phases lysed by method 3 (IGEPAL+freeze-thaw+proteinase K) exhibited lower $C_T$ than the gDNA control. This indicates that more gDNA is extracted by method 3 than by the DNeasy Blood and Tissue Kit from the same amount of cell culture. Thus, method 3 exhibits better results than the DNeasy Blood and Tissue Kit for Gram-negative strains. This is probably because the direct PCR method avoids gDNA loss due to binding on the spin column of the DNeasy Blood and Tissue Kit. Methods 1 (IGEPAL only) and 2 (IGEPAL+freeze-thaw) exhibited similar efficiency for extracting gDNA from Gram-negative bacteria (*Escherichia coli* and *Pseudomonas putida*) in both exponential and stationary phases.

The gDNA of exponential-phase cultures of *Lactococcus lactis* and *Lactobacillus brevis* was as effectively extracted by method 3 as by the gDNA control (DNeasy Blood and Tissue Kit). As shown in Fig. 2 and Table S1, the $\Delta C_T$ of exponential-phase cultures of *Lactococcus lactis* ($\Delta C_T = 0.19 \pm 0.7$ and $P = 0.787$) and *Lactobacillus brevis* ($\Delta C_T = 0.70 \pm 0.5$ and $P = 0.19$) extracted by method 3 is close to that of their gDNA control extracted by the DNeasy Blood and Tissue Kit. Both method 1 and method 2 were less efficient in extracting gDNA of exponential-phase cultures of *Lactococcus lactis* and *Lactobacillus brevis* than the DNeasy Blood and Tissue Kit. For example, the amount of gDNA from exponential-phase cultures of *Lactococcus lactis* and *Lactobacillus brevis* extracted by method 2 was 4.6-fold ($\Delta C_T = -2.56 \pm 0.86$, $P = 0.016$) and 7.6-fold ($\Delta C_T = -3.11 \pm 0.24$, $P < 0.0001$) lower, respectively, than that obtained with the DNeasy Blood and Tissue Kit. In addition, the direct PCR methods had difficulties effectively extracting gDNA from the stationary-phase cultures of both *Lactococcus lactis* and *Lactobacillus brevis*, in particular with method 1 ($\Delta C_T = -3.11 \pm 0.24$ for *Lactococcus lactis* and $\Delta C_T = -3.11 \pm 0.24$ for *Lactobacillus brevis* with $P < 0.0001$ for both). Apparently, the surfactant could not disrupt the thick peptidoglycan in the cell wall of Gram-positive bacteria. Method 3 improved the efficiency significantly by adding both freeze-thaw and proteinase K treatment. As expected, the efficiency of cell disruption increased from method 1 to 3. In Fig. S1B, the qPCR curves of the exponential-phase *Lactococcus lactis* from methods 1 to 3 shift from right to left, getting close to that of the gDNA control. In addition, the cell cultures of exponential-phase *Lactococcus lactis* treated by the direct PCR method from method 1 to 3 became more transparent, as shown in Fig. S1C. To further improve the gDNA extraction of stationary-phase Gram positive bacteria, lysozyme treatment (10 mg/ml lysozyme at 55°C for 20 min) was applied before proteinase K treatment in method 3. In this design, the lysozyme disrupts the peptidoglycan, and proteinase K denatured lysozyme to eliminate the potential inhibitive effects on PCR. However, either lysozyme or proteinase K-denatured lysozyme inhibits PCR (the gel image is not shown.).

**Evaluation of direct PCR methods using a mock microbial community standard.** To evaluate the ability of the direct PCR methods to accurately estimate the relative abundance of members in more complex microbial communities, we tested them against the most widely used microbiome gDNA extraction kit, the DNeasy PowerSoil kit, using a mock community standard, the ZymoBIOMICS microbial community standard. The ZymoBIOMICS microbial community standard is composed of three Gram-negative and five Gram positive bacteria along with two yeast strains (not measured in our study). They differ in GC content, genome size, and 16S copy number and are mixed in a known ratio. The composition of the ZymoBIOMICS microbial community standard was measured by 16S sequencing either following gDNA extraction by PowerSoil and 16S PCR or following direct 16S PCR methods. The relative abundances of the standard community and those measured by PowerSoil and the three direct PCR methods are listed in Table S2.

All methods were able to extract DNA from all microbial species. The average composition of each species of the standard community and those obtained with PowerSoil and the direct PCR methods are visualized in Fig. 3A. At a glance, the relative abundance obtained from method 3 and PowerSoil are much closer to the real composition of the standard community than those from methods 1 and 2. To quantify the similarity of the relative abundances obtained by the direct PCR methods and PowerSoil to the real relative abundance of the standard community, we calculated the Euclidean distances between the relative abundance of each measurements and the real relative abundance of the standard community and considered the Euclidean distances as the index of dissimilarity. In Fig. 3B, the dissimilarity of the composition from method 3 to the composition of the standard community ($0.29 \pm 0.01$) is close to the dissimilarity of the composition from PowerSoil to the composition of the standard community ($0.22 \pm 0.05$), although it is not exactly the same ($P = 0.02$, Welch's *t* test). The dissimilarity between the composition from method 1 and the composition of the standard community and

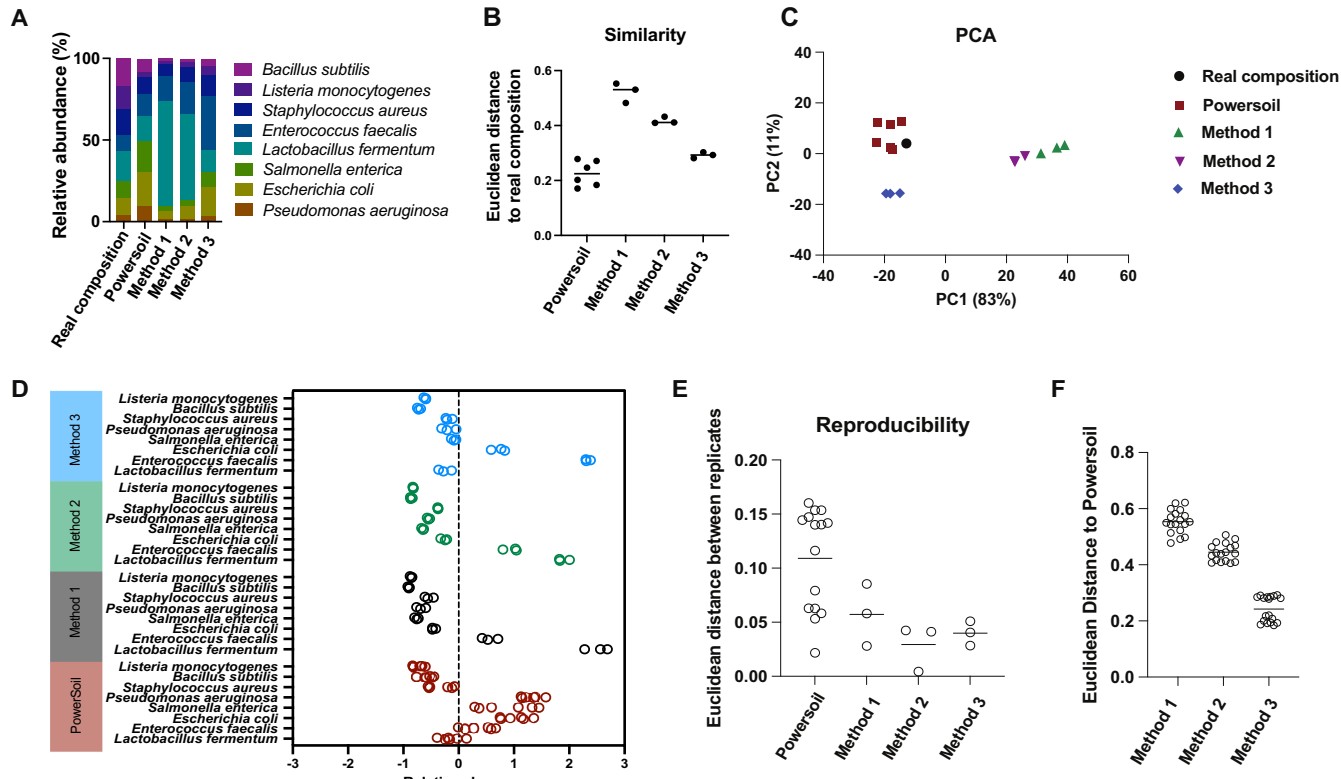

**FIG 3** Evaluation of direct PCR methods and DNeasy PowerSoil Kits using ZymoBIOMICS microbial community standards. (A) Bacterial average composition of ZymoBIOMICS microbial community standards and those obtained by PowerSoil and direct PCR methods. (B) Euclidean distance of the relative abundance obtained from direct PCR methods and PowerSoil to the real relative abundance of the standard community (similarity to the real composition). (C) PCA plot derived from Euclidean distances among the real relative abundance of the standard community, the relative abundance obtained from direct PCR methods, and that from PowerSoil. (D) Relative changes of the relative abundance of each bacterium in ZymoBIOMICS microbial community standards using direct PCR methods and PowerSoil compared to the real relative abundance of the standard community. Positive changes indicate higher relative abundance in the methods compared to the real relative abundance of the standard community, and negative changes indicate lower relative abundance in the methods compared to the real relative abundance of the standard community. (E) Euclidean distances among replicates (repeatability of each method). (F) Euclidean distance of the direct PCR methods to PowerSoil (similarity to PowerSoil).

the dissimilarity between the composition from method 2 and the composition of the standard community are 0.52 ± 0.04 and 0.42 ± 0.01, respectively. Both method 1 and method 2 have a much higher bias than method 3 (0.29 ± 0.01) and PowerSoil (0.22 ± 0.05). Figure 3C shows the principal-component analysis (PCA) plot of the relative abundances obtained from different methods compared to the real relative abundance of the standard community, in which the same symbols represent the replicates of each method. Principal component 1 (PC1), which takes 83% weight, dominates the differences.

To further understand the bias on different bacteria among different methods, the relative changes of the relative abundance of each member by different methods compared to the real relative abundance of the standard community are shown in Fig. 3D. These relative changes of relative abundance compared to the real relative abundance of the standard community are defined as the fold bias [(relative abundance measured − real relative abundance of the standard community)/real relative abundance of the standard community]. Almost all the bacteria showed <1-fold bias in method 3 (IGEPAL+freeze-thaw+proteinase K) compared to the real relative abundance of the standard community, except *E. faecalis*, which had a 2.3-fold overestimation of relative abundance. In particular, the relative abundance biases of *S. aureus*, *P. aeruginosa*, *S. enterica*, and *L. fermentum* measured by method 3 were very small: −0.18 ± 0.07-, −0.18 ± 0.13-, −0.09 ± 0.04-, and −0.25 ± 0.12-fold of the relative abundance of the standard community, respectively. The relative abundances of three Gram-negative bacteria—*E. coli*, *P. aeruginosa*, and *S. enterica*—measured by PowerSoil had a >1-fold positive bias compared to the real relative abundance of the standard community. This demonstrates the

systematic bias of the Gram-negative preference of the PowerSoil DNA extraction kit. The relative changes in relative abundance of almost all the bacteria decreased from method 1 (IGEPAL only) to method 3, except for *E. faecalis*. It suggests that the gDNA extraction efficiency on most of the bacteria improves from method 1 to 3, and *E. faecalis* seems to be the outlier. In addition, the Gram-positive bacterium *L. fermentum*, which has the smallest genome (1.905 Mb) in the mock community standard, had the highest positive bias of relative abundance measured in method 1 compared to the real relative abundance of the standard community, which is much higher than for all the Gram-negative bacteria—*P. aeruginosa*, *E. coli*, and *S. enterica*. This indicates that IGEPAL may be effective enough to disrupt some Gram-positive strains, like *L. fermentum*.

To determine the quantitative repeatability/reproducibility of replicate samples with the direct PCR methods and PowerSoil, we used the Euclidean distances of the composition of each replicate measurement for each method (Fig. 3E). A smaller Euclidean distance indicates higher repeatability. Direct PCR methods 2 and 3 are more reproducible than PowerSoil, since the Euclidean distance scores of method 2 and method 3 ($0.057 \pm 0.028$ and $0.029 \pm 0.022$, respectively) are much lower than that of PowerSoil ($P = 0.003$ for PowerSoil compared to method 2; $P = 0.0002$ for PowerSoil compared to method 3; Welch's *t* test). There are more PowerSoil replicates than in our direct PCR methods because the first set of PowerSoil replicates exhibited low repeatability; to determine whether this was due to the system deviation of the method or the experiment error, we doubled the number of PowerSoil replicates. Also, this is consistent with our results of the first set: the direct PCR is more reproducible than PowerSoil.

Since PowerSoil is widely used in microbiome gDNA extraction, we also compared the dissimilarity of the relative abundance measured by the direct PCR methods (methods 1, 2, and 3) to that of PowerSoil using Euclidean distance as the index. As shown in Fig. 3F, the dissimilarity, defined as the Euclidean distance, to PowerSoil decreases from method 1 to 3 ($P < 0.001$, one-way analysis of variance [ANOVA]). In addition, the dissimilarity of the relative abundance obtained from method 3 to PowerSoil is $0.24 \pm 0.045$, similar to the dissimilarity of the relative abundance from PowerSoil to the real relative abundance of the standard community ($0.22 \pm 0.05$; $P = 0.45$, Welch's *t* test).

**Evaluation of direct PCR methods by analyzing the microbiome in groundwater samples.** To determine the practical outcomes that stem from differences in the above methods of quantifying microbial communities of environmental samples, we tested the direct PCR methods on groundwater samples (GW822D and GW823E) from Bear Creek Valley watershed of Oak Ridge, TN. Since the "real" compositions of the communities of GW822D and GW823E are unknown, we evaluated the microbial composition obtained by the direct PCR methods compared to the microbial composition obtained by PowerSoil, which has been widely used and adopted as the standard protocol by the Earth Microbiome Project (2, 15). The relative abundances of exact sequence variants (ESVs) obtained by the direct PCR methods (methods 1, 2, and 3) and PowerSoil are listed in Fig. S2. The average composition of ESVs of PowerSoil and the direct PCR methods are visualized in Fig. 4A and B. The average composition at the phylum level is shown in Fig. S3. At a glance, the compositions obtained from direct PCR methods are all very similar to that of PowerSoil. The dissimilarity is further quantified by the Bray-Curtis distance of each direct PCR method to PowerSoil using the normalized and log-transformed ESV metrics. The results are shown in Fig. 4C and D. The Bray-Curtis distances of the microbial composition of GW822D obtained from methods 1, 2, and 3 to that of PowerSoil are $0.12 \pm 0.023$, $0.076 \pm 0.028$, and $0.12 \pm 0.010$, respectively. For sample GW823E, the Bray-Curtis distances obtained from methods 1, 2, and 3 to PowerSoil are $0.20 \pm 0.052$, $0.096 \pm 0.031$, and $0.11 \pm 0.033$, respectively. The distance between method 1 and PowerSoil is higher than the distances between method 2 or 3 and PowerSoil ($P < 0.05$ for both, Welch's *t* test).

The principal-coordinate analysis (PCoA) based on the Bray-Curtis distances is shown in Fig. 4E. The dissimilarity is highly dependent on principle component 1 (87%). The data are strongly clustered according to the sample types (GW822D and GW823E), with minor separations among methods. All of the direct PCR methods are

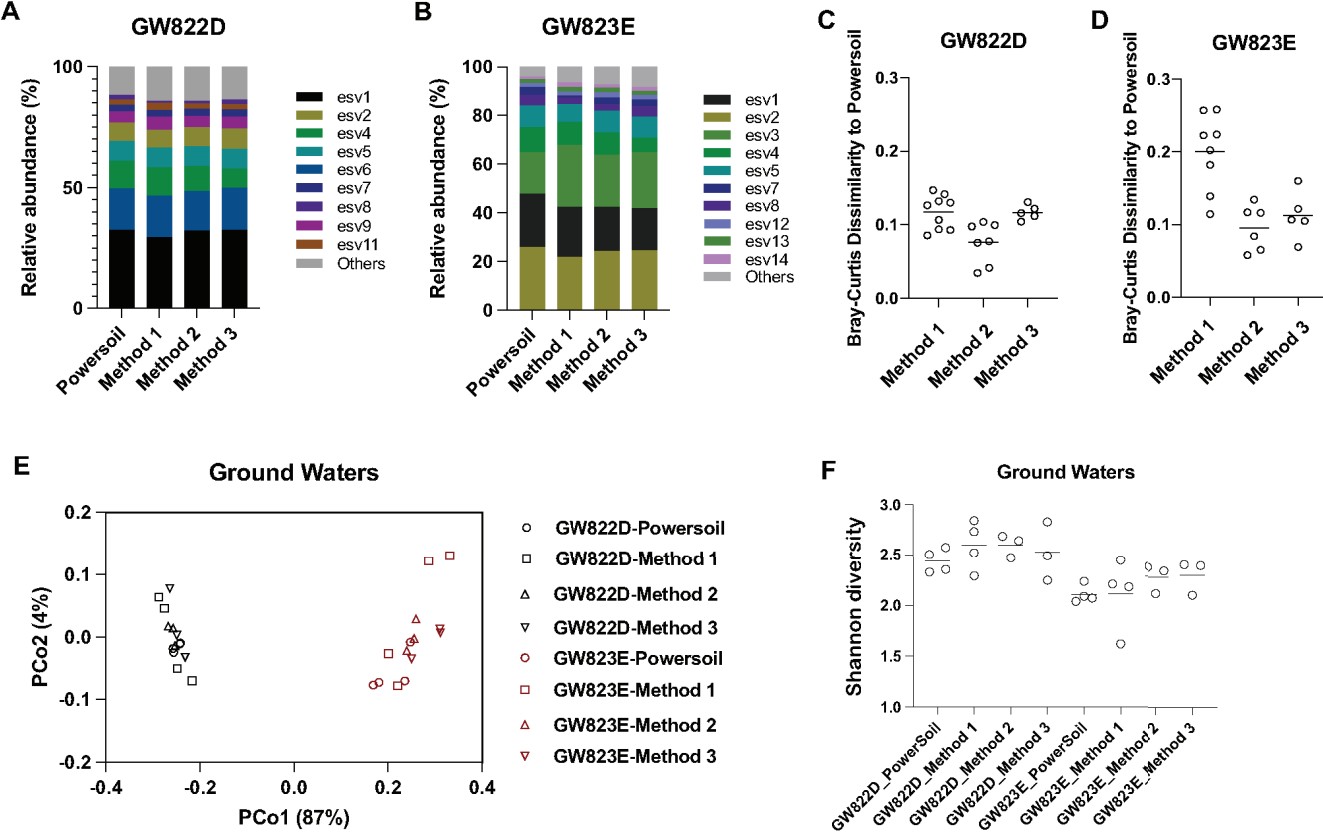

**FIG 4** Comparison of direct PCR methods to PowerSoil kits using groundwater samples. (A) Average bacterial exact sequence variants (ESV) composition of the groundwater sample GW822D. (B) Average bacterial ESV composition of the groundwater sample GW823E. (C) Bray-Curtis distances of the composition of GW822D obtained by direct PCR methods to PowerSoil. (D) Bray-Curtis distances of the composition of GW823E obtained by direct PCR methods to PowerSoil. (E) PCoA plots derived from Bray-Curtis dissimilarities of the community compositions obtained using the direct PCR methods and PowerSoil. (F) Alpha diversity (Shannon index) of GW822D and GW823E obtained by PowerSoil and direct PCR methods.

able to reflect the differences of these two water samples (permutational multivariate analysis of variance [PERMANOVA], $P < 0.001$ by comparing samples). However, method 2 and 3 are closer to the results obtained from the PowerSoil method. We also evaluated the differences in alpha diversity (Shannon index) of samples GW822D and GW823E between the direct PCR and the PowerSoil method, which is shown in Fig. 4F. The alpha diversities are different between samples GW822D and GW823E ($P = 0.002$, Welch's t test), but there are no significant differences across different methods ($P >$ 0.3 for both GW822D and GW823E, one-way ANOVA).

**Some aspects of the new direct PCR methods.** The efficiency and repeatability of the direct PCR methods were evaluated via both testing on the mock microbial community standard and comparing the direct PCR methods against the DNeasy Blood and Tissue DNA extraction kit and the DNeasy PowerSoil DNA extraction kit. Like all the DNA extraction methods, the direct PCR methods have biases. However, the efficiency of the direct PCR methods, in particular method 3, for the samples we tested was comparable to that of the widely used DNA extraction kits, i.e., the DNeasy PowerSoil kit and the DNeasy Blood and Tissue Kit.

The direct PCR methods are more cost-effective, simple, and automation friendly. Table 2 summarizes the cost, complexity, and compatibility of the DNeasy PowerSoil kit, the three direct PCR methods, and the previously described Extract-N-Amp plant PCR kit-based direct PCR method (15, 16), which includes extraction solution addition, heating, and dilution solution addition (15) and is modified by adding a shaking step using TissueLyser prior to using the Extract-N-Amp plant PCR kit (16). The costs of the direct PCR methods, which are $0.004/96 samples for method 1, $0.012/96 samples for

method 2, and $0.34/96 samples for method 3, are all negligible and much lower than the costs of the DNeasy PowerSoil HTP 96 kit ($552.7/96 samples) and the Extract-N-Amp plant PCR kit ($176.9/96 samples). The cost analysis in detail is listed in Table S3. In particular, as the best-performing direct PCR method, method 3 (IGEPAL+freeze-thaw+proteinase K) is 1,600 times less expensive ($0.34/96 samples) than PowerSoil ($552.7/96 samples) and 500 times less expensive than Extract-N-Amp ($176.9/96 samples).

The protocol of direct PCR is also simpler and shorter than PowerSoil (Fig. 1). Method 1 does not require any extra steps except adding IGEPAL to the PCR reagent. Method 2 involves the addition of IGEPAL and a freeze-thaw step. Method 3 adds proteinase K treatment on top of method 2. The Extract-N-Amp plant PCR kit has extraction solution addition, heating treatment, and dilution solution addition steps (15). Videvall et al. also used an additional shaking step using TissueLyser (Qiagen) on top of the Extract-N-Amp plant PCR protocol (16). All three direct PCR methods including Extract-N-Amp are much simpler than the DNeasy PowerSoil HTP 96 kit, which is a 32-step process as described in the manufacturer's protocol. Accordingly, the total time needed to perform any of the direct PCR methods is much shorter than that for PowerSoil. Method 1 requires negligible time. Methods 2 and 3 take 45 min and 2 h, respectively. It is easy to process multiple 96-well plates using these methods, which reduces the processing time per plate. Extract-N-Amp takes 25 min. Also, PowerSoil takes at least 4 h for extracting gDNA from a 96-well plate. Although method 3 takes a total of 2 h, 1 h 45 min is occupied by thermal cycling. The hands-on time is only 15 min. Method 3 is 10 times simpler (3 steps and 15 min hands-on time for 96 samples) than PowerSoil (32 steps and 3.5 h hands-on time for 96 samples). In addition, since the operation in method 3 is either reagent (IGEPAL or proteinase K) addition or thermal cycling, all the steps are compatible with automation.

## DISCUSSION

The accuracy and efficiency of the direct PCR methods increased from method 1 to method 3. In method 3 (IGEPAL+freeze-thaw+proteinase K), IGEPAL and proteinase K disrupt the phospholipids and membrane proteins; freeze-thawing and heating mechanically disrupt the bacterial membrane. The efficiency of method 3 is comparable to that of widely used kits, including the DNeasy Blood and Tissue DNA extraction kit and the DNeasy PowerSoil DNA extraction kit. In particular, method 3 exhibits higher efficiency than the DNeasy Blood and Tissue DNA extraction kit in disrupting Gram-negative bacteria (*Escherichia coli* and *Pseudomonas putida*). Importantly, method 3 is 1,600 times less expensive and 10 times faster to execute (in terms of hand-on time) than the DNeasy PowerSoil DNA extraction kit. The direct PCR methods are also compatible with automation and miniaturization. Because method 3 involves only IGEPAL and proteinase K addition, freeze-thawing, and thermal cycling, the protocol could be easily integrated into automated laboratory robots. Also, because high-throughput cultivation assays of microbial communities tend to be performed using smaller volumes, such as with droplets or in 1,536-well plates, current column-based or bead-based DNA extraction methods will be incapable of processing these kinds of samples due to the significant DNA loss for tiny-volume and low-cell-density samples (7–10, 24). The direct PCR method will solve this problem by keeping all the gDNA in the PCR without any loss. If the efficiency and precision were the same, the choice of methods would be guided largely by the cost, time, and potential for automation (25). Thus, the direct PCR method is a promising approach to microbial community analysis in this circumstance.

Achieving perfect quantification of a bacterial community to unbiasedly reflect the true composition is extremely difficult, even for the popular and conventional PowerSoil kit. Reasons include different bacteria releasing gDNA on different time scales (due to the complexity of bacterial structures), causing the gDNA that is released early in the process to be at risk of being degraded; certain bacterial types being difficult to lyse; and bias via PCR amplification, sequencing, and bioinformatics (5). Unsurprisingly, the direct

PCR method 3 has bias, too. The bacterial disruption bias of the direct PCR method 3 is likely due to the lack of specific peptidoglycan disruption reagents, such as lysozyme. Because mechanical disruption such as freeze-thawing could theoretically disrupt peptidoglycan, method 3 has difficulties only in disrupting bacteria with very thick peptidoglycan, such as late-stationary-phase Gram-positive bacteria, which is consistent with our results; the qPCR results demonstrate low efficiency of the direct PCR methods for disrupting stationary-phase *Lactococcus lactis* and *Lactobacillus brevis* but similar efficiency in disrupting exponential-phase *Lactococcus lactis* and *Lactobacillus brevis* compared with the DNeasy Blood and Tissue DNA extraction kit. To improve the disruption of peptidoglycan, an enzyme treatment followed by the proteinase K denaturation of this enzyme is a promising solution. Although denatured lysozyme has been shown to inhibit PCR, determining the reason for the inhibition would help with designing lysozyme treatment in the direct PCR. In addition, identifying other enzymes that do not influence PCR would be another way to improve the direct PCR method. Also, since the quantification bias could be caused not only by DNA extraction but also by PCR amplification, sequencing, and bioinformatics (5), it is important to systematically improve all the steps to better estimate absolute quantification.

The existence of PCR inhibitors in various samples is the limitation of this direct PCR method. Since the chemical composition of microbiome samples is extremely diverse, it is impossible to have a general protocol fitting all the samples. Additional treatment may be required based on our direct PCR method 3. For example, to process the samples with low pH, high-ion samples, or samples with a soluble PCR inhibitor such as glycerol, it is necessary to remove the liquid phase by centrifuging to pellet the sample. However, in the case of the rich and complex soil samples with high humic acid and unknown PCR inhibitors, DNA extraction-based methods such as PowerSoil are recommended.

One of the potential applications of the direct PCR method is to measure the growth curves of all the species in microbial communities. The recent advances in sequencing-based quantification have extended its ability to obtain relative bacterial abundance to make quantifying the absolute bacterial abundance possible; by adding reference genes (26, 27), measuring total gene load (28), or using new algorithms (29), it provides a unique way to quantify the bacterial growth curve and thereby assess the bacterial interactions and assembly in complex bacterial communities using genetic markers such as 16S genes (30). Due to advances in sequencing-based quantification and the decrease of the cost of sequencing, the sequencing-based method will be a promising alternative to optical density ($OD_{600}$)-based or fluorescence-based growth curve measurements to quantify the microbial growth in a community. One of the advantages of sequencing-based quantification is the ability to target the growth of multiple strains simultaneously. However, it is still technically very challenging to understand bacterial interactions and assembly in complex microbial communities, since the sample number increases dramatically as the numbers of species, time points, replicates, and initial conditions increase. For example, to understand the interactions and assembly of a 7-member community, testing requires 127 samples with all the combinations of each species for just one time point without replicates. The number increases to 255 for an 8-member community and will explode upon the addition of more time points, more replicates, and various initial ratios of each species. As DNA sequencing is getting less expensive, the bottleneck of this sequencing-based method is the cost of the DNA extraction and purification. Our low-cost, automation-friendly direct PCR assay is able to handle increasingly large numbers of microbial community samples, allowing us to study dynamics, microbial interactions, and assembly in a high-throughput fashion.

Our direct PCR methods have been evaluated by lab microbial enrichment, microbial community standard culture, and environmental water samples. Since there are diverse microbiome samples with unique chemical compositions, we were not able to screen all types of microbiome samples. Thus, we recommend a PCR/qPCR evaluation to check the efficiency of the direct PCR method on particular types of samples we did

not cover. For example, our direct PCR method (method 3) has been shown to successfully extract gDNA from 10-day *E. coli* biofilm samples (Fig. S4).

In summary, we provide a cost-effective, simple, and automation friendly direct PCR method for high-throughput microbial community composition analysis. This successfully demonstrates the possibility of using direct PCR methods on microbial community analysis. Direct PCR method 3 (IGEPAL+freeze-thaw+proteinase K), which is comparable to the widely used commercial kits, has low cost, and is quick to perform, could dramatically increase the throughput of recent 16S sequencing profiling, as well as providing an alternative way to study microbial community assembly and interactions.

## MATERIALS AND METHODS

**Strains and cultivation conditions.** The cell lysis efficiency of direct PCR methods was compared to that of the Qiagen DNeasy Blood and Tissue Kit using the following strains in both stationary phase and exponential phase: *Escherichia coli* strain K-12 substrain MG1655 (referred to here as *E. coli*), *Pseudomonas putida* KT2440 (referred to here as *P. putida*), *Lactococcus lactis* subsp. *cremoris* strain MG1363 (referred to here as *L. lactis*), and *Lactobacillus brevis* ATCC 14869 (referred to here as *L. brevis*). For the stationary-phase cultures, *E. coli* and *P. putida* were inoculated from glycerol stock into LB Lennox medium and cultivated at 37°C with 200 rpm overnight. *L. lactis* and *L. brevis* were inoculated from glycerol stock in MRS medium and cultivated at 30°C without shaking. The $OD_{600}$ of the overnight cultures of *E. coli*, *P. putida*, *L. lactis*, and *L. brevis* were 2.33, 2.28, 1.77, and 3.68, respectively. For exponential-phase cultures, overnight cultures were diluted (1:100) from overnight cultures. Culture conditions for all the strains remained the same. Cultures were collected between 4 and 6 h later. The $OD_{600}$ of exponential-phase cultures of *E. coli*, *P. putida*, *L. lactis*, and *L. brevis* were 0.86, 0.75, 0.39, and 0.16, respectively. Pellets were stored at −80°C until genomic DNA extraction.

**Mock microbial community standard.** ZymoBIOMICS microbial community standards (Zymo Research) were used in this study to quantitatively evaluate the efficiency of direct PCR methods and the DNeasy PowerSoil kit. This microbial community standard is a defined composition comprising 5 Gram-positive bacteria (*Listeria monocytogenes*, *Bacillus subtilis*, *Lactobacillus fermentum*, *Enterococcus faecalis*, and *Staphylococcus aureus*), 3 Gram-negative bacteria (*Pseudomonas aeruginosa*, *Escherichia coli*, and *Salmonella enterica*), and 2 yeast strains (*Saccharomyces cerevisiae* and *Cryptococcus neoformans*). The defined composition is well controlled and reported by the manufacturer as the real composition of the community. The ZymoBIOMICS microbial community standards were stored at −80°C until use. To avoid the effects of glycerol on microbial community standard cell lysis, 40 $\mu$l of the standard was thawed and centrifuged at 10,000 × *g* for 5 min. The supernatant was discarded, and the pellet was resuspended in 80 $\mu$l Milli-Q DNase-free water immediately before direct PCR or gDNA extraction.

**Environmental groundwater samples.** Environmental groundwater samples (GW822D and GW823E) were collected in April 2019 from two uncontaminated background-area wells in Bear Creek Valley watershed of Oak Ridge, TN. Groundwater (50 ml) was vacuum filtered over 0.2 $\mu$m-pore-size filters, and the concentrated microbial community was resuspended in 10 ml phosphate-buffered saline (PBS). This suspension was used for direct PCR or genomic DNA extraction by the DNeasy PowerSoil kit.

**Direct PCR methods.** We tested three variations of direct PCR methods using the detergent IGEPAL as the core lysis reagent. In the IGEPAL-only method (method 1), IGEPAL detergent was added to PCR (0.1% final concentration) with an initial activation set at 98°C for 10 min, the same as for many hot-start DNA polymerases. In the IGEPAL+freeze-thaw method (method 2), PCR template was prepared by adding an equal volume of bacterial cell suspension to IGEPAL (0.5% final concentration) in a PCR plate; samples were mixed 10 times by pipetting up and down, followed by five cycles of freeze-thawing by placing the plates at −80°C for 15 min and then allowing them to thaw at room temperature for 15 min. In the IGEPAL+freeze-thaw+proteinase K method (method 3), samples were prepared in the same way as in the IGEPAL+freeze-thaw method, except that proteinase K (20 mg/ml; Qiagen) was added to the samples after the freeze-thaw cycles in 96-well PCR plates to reach a final concentration of 100 $\mu$g/ml. The plates were sealed and centrifuged at 300 × *g* for 1 min to collect the liquid to the bottom of each well. The samples were treated at 60°C for 1 h, and then enzyme was deactivated at 95°C for 15 min in a thermocycler (Bio-Rad). After the treatment, the sample was ready to use as a DNA template in the subsequent PCR.

**Genomic DNA extraction methods.** Two conventional genomic DNA extraction methods were used in this study for comparison with the direct PCR methods: the DNeasy Blood and Tissue Kit (Qiagen) and the DNeasy PowerSoil kit (Qiagen). The DNeasy Blood and Tissue Kit was used to extract genomic DNA from the four model strains per the manufacturer's specifications, with additional lysozyme treatment for Gram-positive strains. For the gDNA extraction, 1 ml of culture volume was used for the exponential-phase cultures and 400 $\mu$l of culture volume was used for stationary-phase cultures. Samples were eluted in 400 $\mu$l water and stored at −20°C. The bacterial cultures used for gDNA extraction were aliquoted and stored at −80°C. We used DNeasy PowerSoil kits to extract genomic DNA from ZymoBIOMICS microbial community standards and microbes from groundwater samples per the manufacturer's specifications. Three replicates were done for each sample.

**qPCR.** Quantitative PCR (qPCR) was used for comparing the efficiency of the direct PCR methods with the DNeasy Blood and Tissue Kit. The extracted gDNA was from the sample which was used for direct PCR. The gDNA was extracted by the DNeasy Blood and Tissue Kit from 400 $\mu$l stationary-phase culture and eluted with 400 $\mu$l elution buffer, so 1 $\mu$l extracted gDNA was equivalent to 1 $\mu$l original stationary-phase

culture. For the exponential-phase cultures, because the gDNA was extracted from 1 ml cultures and eluted by 400 $\mu$l elution buffer, 1 $\mu$l extracted gDNA was equivalent to 2.5 $\mu$l original exponential-phase culture (1 ml culture volume/400 $\mu$l elution volume = 2.5 $\mu$l). The extracted gDNA concentrations were quantified by the Quant-iT double-stranded DNA (dsDNA) assay kit. Each qPCR was conducted in a 20-$\mu$l reaction mixture with 10 $\mu$l Sso advanced universal SYBR green Supermix (2×), 1.5 $\mu$l primer 534F (5 mM), 1.5 $\mu$l primer 783R (5 mM), 0.4 $\mu$l RNase A (100 mg/ml, Qiagen), 2 $\mu$l 1% IGEPAL CA-630, and either 1 $\mu$l extracted gDNA or equivalent cell cultures pretreated by the direct PCR methods. Each condition was replicated at least three times. The qPCR was performed using a Bio-Rad CFX96 real-time PCR machine with denaturation at 98°C for 10 min, followed by 38 cycles of denaturation at 98°C for 30 s and annealing/elongation at 60°C for 1 min 30 s. The melting curve was tested from 60°C to 95°C with 0.5°C/cycle with increments of 5 s per cycle. The $C_T$ was calculated by the linear regression method.

**16S amplicon PCR and sequencing.** The community structure of the ZymoBIOMICS microbial community standard and two environmental groundwater samples were measured using 16S V3-V4 region Illumina amplicon sequencing, using the DNeasy PowerSoil kit, which is a widely used traditional DNA extraction kit, and the direct PCR methods. Primers used in the 16S amplicon PCR were constructed with TruSeq Illumina adapters, barcodes, phasing, and linker sequences, with 341F and 806R targeting the 16S V3-V4 hyper variable region of the 16S gene, adopted from the work of Justice et al. (31). Genomic DNA extracted from the PowerSoil kit was quantified by the Quant-iT dsDNA assay kit. The concentration of gDNA in ZymoBIOMICS microbial community standards was between 5.4 ng/$\mu$l and 6.0 ng/$\mu$l. The concentration of gDNA in water samples GW822D and GW823E was between 0.31 ng/$\mu$l and 0.38 ng/$\mu$l. The template for the 16S amplicon reaction was either the extracted gDNA from the PowerSoil kit or direct PCR method-treated cells. KAPA HiFi HotStart ReadyMix was used in the PCR. The PCR was conducted in a 30-$\mu$l reaction mixture with 15 $\mu$l KAPA HiFi HotStart ReadyMix (2×), 3 $\mu$l 1% IGEPAL CA-630, 0.6 $\mu$l RNase A (100 mg/ml; Qiagen), 3 $\mu$l forward- and reverse-primer mix (2.5 $\mu$M for each), and 8.4 $\mu$l template (either extracted gDNA or equivalent cells treated by direct PCR methods). (We recommend using KAPA HiFi HotStart polymerase, since we found that either RNase A or denatured RNase A inhibited PCR with Invitrogen Platinum polymerase but not KAPA HiFi HotStart polymerase. If a modified protocol without RNase A is used, Invitrogen Platinum polymerase may be used.) The cycling conditions were 98°C for 10 min, followed by 26 cycles of 98°C for 20 s, 53°C for 30 s, and 72°C for 2 min, and a final extension step of 72°C for 5 min. The PCR products were run on a 1.2% agarose gel to determine the amplicon concentration. Based on the quantification from gel imaging, similar amounts of PCR products of each PCR were pooled and purified with AMPure XP beads per the manufacturer's protocol, and the purified product was quantified by the Quant-iT dsDNA high-sensitivity assay. Then, 4 nM library stock was prepared by diluting the purified amplicon DNA with water. The library stock was denatured and diluted following the manufacturer's instructions; a final concentration of 20 pM denatured library was loaded onto the flow cell and sequenced using a custom read 2 primer (5′- CGGTCTCGGCATTCCTGCTGAACCGCTCTTCCGATCT). Amplicons were sequenced using the Illumina 600-bp v3 kit with 350 bp read 1 and 250 bp read 2 on the Illumina MiSeq platform for a better overall read quality than the previous 2 × 300-bp reads, because the read quality drops dramatically in read 2.

**Sequencing data processing and analysis.** All the analyses were performed in R (3.6.0). The amplicon sequence data were analyzed using DADA2 (32). Most forward and reverse reads did not meet the criteria for high-quality assembly during read merging (less than 1 mismatch and 20-bp overlap-find, actual parameter used for assembly) due to the low-quality sequence of the overlapping region in read 2, so the DADA2 was analyzed by read 1 only, which includes regions V3 and V4. Primer sequences were trimmed with cutadapt, sequence length was trimmed to 300 bp, low-quality reads were filtered by the settings (maxN = 0, maxEE = 3, truncQ = 2), chimeric reads were removed, and the relative abundance of exact sequence variants (ESVs) was calculated by DADA2. The taxonomy was assigned using the naive Bayesian classifier to assign taxonomy across multiple ranks with the SILVA database V132 as a reference (33). ZymoBIOMICS microbial community standard strains were counted by the reads of each species. For the water samples, alpha diversity was calculated based on ESVs using the Shannon index. Bray-Curtis distances was calculated using the normalized and log-transformed ESV metrics and examined with PERMANOVA by using the Adonis function with 1,000 permutations, both in vegan (v2.5-7). In addition, PCoA based on Bray-Curtis distances was used to visualize and compare the different water samples and different methods.

**Data availability.** The compositions of the microbial community standard samples and water samples are provided in Table S2 and Fig. S2. The raw sequencing data are available at Figshare (https://doi.org/10.6084/m9.figshare.14635458). The code is available on GitHub (https://github.com/fasong/Direct_PCR_4_bacterial_community_analysis.git).

## SUPPLEMENTAL MATERIAL

Supplemental material is available online only.

**FIG S1**, TIF file, 0.4 MB.
**FIG S2**, TIF file, 1 MB.
**FIG S3**, TIF file, 0.2 MB.
**FIG S4**, TIF file, 0.6 MB.
**TABLE S1**, PDF file, 0.04 MB.
**TABLE S2**, PDF file, 0.02 MB.
**TABLE S3**, PDF file, 0.1 MB.

## ACKNOWLEDGMENTS

We thank Dan Williams, Kenneth Lowe, Terry Hazen, Dominique Joyner, Katie Walker, Andrew Putt, Regina Wilpiszeski, and Emma Dixon at Oak Ridge National Laboratory for the ENIGMA Spring Sampling to provide water samples GW822D and GW823E. We thank Yolanda Huang and Sean Carim for assembly of our strain standards and helpful discussion.

This work, conducted by ENIGMA (Ecosystems and Networks Integrated with Genes and Molecular Assemblies [https://enigma.lbl.gov/]), a Scientific Focus Area Program at Lawrence Berkeley National Laboratory, was supported by the Office of Science, Office of Biological and Environmental Research, of the U.S. Department of Energy under contract DE-AC02-05CH1123.

F.S. conceived the project and wrote the original draft. F.S. and A.C. conducted the experiments. F.S. and J.V.K. designed the computational framework and analyzed the data. A.P.A. supervised the project. F.S. wrote the manuscript. All authors reviewed and revised the manuscript.

We declare no competing financial interests.

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
