## [Reviewer comments · mSystems]

A simple, cost-effective and automation-friendly direct PCR approach for bacterial community analysis

Fangchao Song, Jennifer Kuehl, Arjun Chandran, and Adam Arkin

Corresponding Author(s): Fangchao Song, Lawrence Berkeley National Laboratory

Review Timeline:

Submission Date:	February 26, 2021
Editorial Decision:	March 22, 2021
Revision Received:	June 4, 2021
Editorial Decision:	July 6, 2021
Revision Received:	August 19, 2021
Accepted:	August 30, 2021

Editor: Sarah Glaven

Reviewer(s): The reviewers have opted to remain anonymous.

Transaction Report:

DOI: <https://doi.org/10.1128/mSystems.00224-21>

March 22, 2021

Dr. Fangchao Song
Lawrence Berkeley National Laboratory
Berkeley

Re: mSystems00224-21 (A simple, cost-effective and automation-friendly direct PCR approach for bacterial community analysis)

Dear Dr. Fangchao Song:

Below you will find the comments of the reviewers.

To submit your modified manuscript, log onto the eJP submission site at <https://msystems.msubmit.net/cgi-bin/main.plex>. If you cannot remember your password, click the "Can't remember your password?" link and follow the instructions on the screen. Go to Author Tasks and click the appropriate manuscript title to begin the resubmission process. The information that you entered when you first submitted the paper will be displayed. Please update the information as necessary. Provide (1) point-by-point responses to the issues raised by the reviewers as file type "Response to Reviewers," not in your cover letter, and (2) a PDF file that indicates the changes from the original submission (by highlighting or underlining the changes) as file type "Marked Up Manuscript - For Review Only."

Due to the SARS-CoV-2 pandemic, our typical 60 day deadline for revisions will not be applied. I hope that you will be able to submit a revised manuscript soon, but want to reassure you that the journal will be flexible in terms of timing, particularly if experimental revisions are needed. When you are ready to resubmit, please know that our staff and Editors are working remotely and handling submissions without delay. If you do not wish to modify the manuscript and prefer to submit it to another journal, please notify me of your decision immediately so that the manuscript may be formally withdrawn from consideration by mSystems.

Sincerely,

Sarah Glaven

Editor, mSystems

Journals Department
Reviewer comments:

Reviewer #1 (Comments for the Author):

A very nice, convincing paper that shows the utility and potential of direct PCR methods for high-throughput quantifications and characterizations of microbial communities. Until this method works for complex/inhibitor-rich natural samples more adaptations will be needed, but it is already quite remarkable how well it works on pure cultures during different growth stages, mock communities, and groundwater samples.

I only have two very minor comments:

L. 179-194: Please explain how the abundance measured by qPCR can exceed the actual abundance.

L. 271-272: Please provide ideas how cells with thick peptidoglycan cell walls might be more effectively lysed.

I also noticed some minor grammatical errors, especially in the Discussion. These can be fixed by a native speaker of English.

Reviewer #2 (Comments for the Author):

In this study by Song et al., they compare three "direct PCR" methods with two DNA extraction methods (DNeasy Blood and Tissue and DNeasy Powersoil). These three direct PCR methods are: (i) the addition of the surfactant IGEPAL; (ii) the addition of IGEPAL and freeze-thaw cycles; and (iii) the addition of IGEPAL, freeze-thaw cycles and the addition of proteinase K and heating cycles. They first use pure cultures of two gram positive and two gram negative bacterial species in either stationary or exponential phase to assess the efficiency of detecting the 16S rRNA genes of these isolates with qPCR when one of their three methods or the Blood and Tissue Kit is used. They then use the Zymobionics mock community as well as two environmental groundwater samples to test the Powersoil kit against their three direct PCR methods, performing Illumina MiSeq sequencing to determine the differences in composition in comparison with the expected composition for the mock community and to each other for both. They show similar extraction efficiency for some strains with the qPCR method (between the Blood and Tissue and Method 3) and that the Powersoil kit better captures the expected composition of the mock community than any of their methods. While I do agree that it is important to test out whether different - and in particular simpler and cheaper - methods can be used given how much microbial community sequencing is currently being performed, I don't feel that the authors have adequately persuaded me that this method

would be widely applicable, particularly as they do not test biofilm samples, or any samples where the cells are not planktonic, e.g., soil, other organic matter or faeces. I also think that some of the conclusions drawn by the authors are not supported by the results and was confused by some of their methodological choices. Below I have detailed my major and minor concerns.

Major comments

- The authors do not go into detail on which kinds of bacterial communities these direct PCR methods may be appropriate for, and I feel that in particular biofilm samples are likely to have a very different efficiency with little or no mechanical or enzymatic disruption. They have only a single sentence suggesting that the methods may not be appropriate for soil due to potential inhibitors such as humic acid (that is right at the end of the discussion). It was disappointing to me that samples from other commonly-samples environments were not included in this analysis, and I think it needs to be made clear throughout that the authors can't make recommendations on bacterial communities generally, only the planktonic ones that they actually test.
- I didn't understand why the authors chose to use the Blood and Tissue kit for their qPCR evaluation of the pure cultures and the Powersoil kit for the community analyses. I would have ideally liked to see both of them used for both analyses, but at a minimum think they should have both been used for the qPCR evaluation and then the better performing one used for the community analyses. It also would have been good if kits from other manufacturers had been included, as the Qiagen kits tested are among the most expensive but are not always found to perform best.
- In the discussion the authors discuss the same points over and over, repeating themselves many times, but do not devote any space to discussing the limitations of this method or any other alternatives.
- The authors should provide the data from sequencing runs (deposited in e.g. NCBI SRA) as well as data and scripts used for analysis.

Minor comments

- Line 16: amount -> number
- Line 18: "biased on the types of species" is incomplete
- Line 49: what outcome?
- Throughout: Please use 16S (capital 'S') consistently
- Line 68-69: I agree that this information can be difficult to find, but it is not impossible and all kits follow a relatively similar formulation so I disagree with this statement.
- Lines 78-88: Please include the Blood & Tissue kit in this description.
- Line 325: in the -80 -> at -80
- Line 332: standards was -> standard was or standards were
- Line 340: Methods elsewhere are specific about volumes used, but they are missing in this paragraph.
- Line 349: 300 g -> 300 x g?
- Line 352: Genome DNA -> Genomic DNA
- Line 361-366: I found this paragraph very hard to follow. Please restructure/rewrite. It would also be useful if the qPCR and MiSeq steps (as well as use of different pure cultures/communities) were added to Figure 1 so that it is easier to see how these both fit in.
- Line 394: Are the read lengths given here correct? Why did they deviate from the standard 2x300?
- Line 397: Is this referring to mismatches in the primer sequences? Please be specific.
- Line 108-113: Why are these other primers not also included in the methods? There are also no details of what was used as a standard curve for the qPCR and therefore of how qPCR efficiency was checked and the values were normalised.

- Figure 2: I initially found this very difficult to understand. I think it would drastically be improved by not showing the Blood & Tissue results 3 times, but rather grouping by sample so that the results of each different method can easily be seen alongside one another. It also would be useful to show the change in Ct values between Blood & Tissue and the direct PCR methods as a separate panel. Please also ensure that axis text labels are sufficiently separate from one another to be clear on any revised figure.
- Lines 127-136: I found it difficult to assess whether I agree with the interpretation of the authors here, but I disagree with the statement that they performed similar to or better than the kit - the only case where this is true is for the gram negative with Method 3. The other methods are similar in some cases, but also worse in many.
- Line 131-132: Please specify that this is only for gram negative.
- Paragraph starting on line 156: I am not sure why the authors chose to use Euclidean distance here, especially as they later use Bray-Curtis distance. I think that the Bray-Curtis distance is the more appropriate to use here, and they should use this instead.
- Line 182: I'm not really clear on why this fold bias has been used rather than just log₂ fold change or something similar that is typically used to look at differences in abundance between samples.
- Lines 195-200: Please rephrase as this part was hard to understand what was meant.
- Figure 3: Why are there more powersoil replicates than direct PCR? I didn't think differences in number of replicates were discussed in the methods. It would also be nice to add the mean differences (both for each species and overall) to panel D and please clarify on how panel E was calculated - is it all vs all, or is it average of all others against the replicate in question.
- Figure 4: please plot ESVs in the same order between the two samples, this makes it quite difficult to look at.
- Colours in stacked bars for both Figures 3 and 4 are probably not colour blind friendly.
- Line 217: There was no mention of normalisation/log transformation for the data shown in Fig. 3, so why is it done now? Consistency between the analyses would be good.
- It would be good to give results of Adonis tests on community compositions.
- Line 244-257: Please rephrase and reduce this paragraph. Much of it is repeated.
- Line 252: Why mention Extract-N-Amp in the results and in Table 2? It was not compared here. Table 2 should also have Blood & Tissue added.
- Line 264: Replace accuracy with efficiency
- Line 278: difficult lyse -> difficult to lyse
- Line 279-280: I do not agree at all with this statement. This is not what is shown by their results.
- Lines 285-296: I have never seen an experiment set up in the way described here, where each combination of species in a system is grown and sequenced. While I agree here that the time of the DNA extractions for this suggested experiment would be a lot, I suspect that the real issue stopping an experiment like this is the sequencing cost (much more than DNA extraction).
- Lines 290-293: while there are some studies suggesting absolute quantification of microbial abundances, these are contested within the microbiome community and if the authors want to make statements like this then the other side must also be discussed. Typically, sequencing is not accepted as absolute quantification. Even with the inclusion of other marker genes or qPCR, copy number can vary drastically between even closely related species.
- Lines 300-301: There is a large assumption here that all of the DNA in a cell will be available for PCR, and I am not convinced this is true with these methods.

Reviewer #1:

A very nice, convincing paper that shows the utility and potential of direct PCR methods for high-throughput quantifications and characterizations of microbial communities. Until this method works for complex/inhibitor-rich natural samples more adaptations will be needed, but it is already quite remarkable how well it works on pure cultures during different growth stages, mock communities, and groundwater samples.

We really appreciate the time and comments of Reviewer 1. We totally agree that a more adaptable direct PCR method will be fantastic. Since the microbiome and their environment is very diverse, it is challenging to have an universal method that can tolerate all the possible inhibitors without DNA purification, particularly in inhibitor-rich soil samples. Based on our direct PCR method, it will be a nice future work to discover case specific strategies to eliminate the effects of particular inhibitors on direct PCR. We have updated our discussion (L. 289) in the revised manuscript.

I only have two very minor comments:

L. 179-194: Please explain how the abundance measured by qPCR can exceed the actual abundance.

We compared the amount of genomic DNA released by direct PCR to the DNeasy Blood and Tissue kit based on qPCR Cts, this is not related to L. 179-194 (L178-194 in the revised manuscript). What I believe reviewer 1 meant to say is “how can the abundance measured by direct PCR exceed the actual abundance”, since in L. 178-194 we compared the relative abundance from our direct PCR method with the actual relative abundance. There is a miscommunication. We corrected “abundance” to “relative abundance” accordingly throughout the manuscript. Relative abundance values can reflect either high DNA extraction/PCR efficiency of the target species or low DNA extraction/PCR efficiency of other species that would cause the target’s relative abundance (composition) higher than the actual relative abundance (composition).

L. 271-272: Please provide ideas how cells with thick peptidoglycan cell walls might be more effectively lysed.

This is something we already touched and planned to work on in the future. Enzyme treatment would be a potential solution for lysing cells with thick peptidoglycan cell walls. However, simply adding lysozyme in the direct PCR would cause gelation during heating. We tried treating our sample with lysozyme first, and then denatured the lysozyme with proteinase K before PCR. We also considered the potential PCR inhibition by the glycerol in the commercial lysozyme solution, and made our own lysozyme solution from the powder. However, PCRs were inhibited in all the cases. Understanding why denatured lysozyme inhibits PCR would help with the design of lysozyme treatments for direct PCR. In addition, looking for other enzymes that don’t inhibit PCR after being denatured by proteinase K would be another way. We have updated the relevant paragraph (L.149 and L.283).

I also noticed some minor grammatical errors, especially in the Discussion. These can be fixed by a native speaker of English.

We have rewritten the whole discussion, and corrected the grammatical errors accordingly in the whole manuscript.

Reviewer #2:

In this study by Song et al., they compare three "direct PCR" methods with two DNA extraction methods (DNeasy Blood and Tissue and DNeasy Powersoil). These three direct PCR methods are: (i) the addition of the surfactant IGEPAL; (ii) the addition of IGEPAL and freeze-thaw cycles; and (iii) the addition of IGEPAL, freeze-thaw cycles and the addition of proteinase K and heating cycles. They first use pure cultures of two gram positive and two gram

negative bacterial species in either stationary or exponential phase to assess the efficiency of detecting the 16S rRNA genes of these isolates with qPCR when one of their three methods or the Blood and Tissue Kit is used. They then use the Zymobiomics mock community as well as two environmental groundwater samples to test the Powersoil kit against their three direct PCR methods, performing Illumina MiSeq sequencing to determine the differences in composition in comparison with the expected composition for the mock community and to each other for both. They show similar extraction efficiency for some strains with the qPCR method (between the Blood and Tissue and Method 3) and that the Powersoil kit better captures the expected composition of the mock community than any of their methods. While I do agree that it is important to test out whether different - and in particular simpler and cheaper - methods can be used given how much microbial community sequencing is currently being performed, I don't feel that the authors have adequately persuaded me that this method would be widely applicable, particularly as they do not test biofilm samples, or any samples where the cells are not planktonic, e.g., soil, other organic matter or faeces.

We really appreciate the time and comments of Reviewer 2. This is an accurate summary of our manuscript. During the revision, we have updated the manuscript accordingly, including a result about biofilm (L.310). Here, we present a point-by-point to reviewers' comments.

I also think that some of the conclusions drawn by the authors are not supported by the results and was confused by some of their methodological choices. Below I have detailed my major and minor concerns.

I have answered the comments below, but want to emphasize here on (1) why we choose Blood and Tissue kit for pure cultures and Powersoil for microbial community samples; (2) why we choose Euclidean distance for microbial community standard but Bray-Curtis dissimilarity for water samples. Briefly, we want to choose the most appropriate kit and method in each case as the standard for our efficiency evaluation. Blood and Tissue is optimized for pure cultures, and Powersoil is designed for environmental samples. Euclidean distance is the true distance between two samples, which is always the first choice to analyze the similarity. But for the samples with many zero or low abundant species, Bray-Curtis is better since it can limit the influence of small numbers on the results. The real composition of our microbial community standard is known to be non-zero, so we choose the Euclidean distance. Water samples contain many low abundant/zero abundant species, thus Bray-Curtis dissimilarity is chosen in that case.

Major comments

• The authors do not go into detail on which kinds of bacterial communities these direct PCR methods may be appropriate for, and I feel that in particular biofilm samples are likely to have a very different efficiency with little or no mechanical or enzymatic disruption. They have only a single sentence suggesting that the methods may not be appropriate for soil due to potential inhibitors such as humic acid (that is right at the end of the discussion). It was disappointing to me that samples from other commonly-samples environments were not included in this analysis, and I think it needs to be made clear throughout that the authors can't make recommendations on bacterial communities generally, only the planktonic ones that they actually test.

This manuscript focuses on liquid cultures including lab enrichments and water samples, and demonstrates the possibility of using direct PCR on diverse microbiome samples. Since the chemical composition of environmental samples are extremely diverse – even soil samples themselves are different from each other, we cannot screen all the possible samples in this manuscript. It would be always good to do a PCR/qPCR to check the efficiency of the direct PCR method on particular types of samples. Regarding biofilms, we did PCR to check the feasibility of direct PCR on 10-days *E. coli* biofilm samples qualitatively. The gel imaging shows the direct PCR works well, indicating the general biofilm matrix would not be a problem for our direct PCR method. In addition, our direct PCR method has been adopted to study human feces by another group, and works very well on their samples. We have updated our manuscript by adding this discussion accordingly (L. 289-294 ; L. 307-311).

• I didn't understand why the authors chose to use the Blood and Tissue kit for their qPCR evaluation of the pure cultures and the Powersoil kit for the community analyses. I would have ideally liked to see both of them used for both analyses, but at a minimum think they should have both been used for the qPCR evaluation and then the better performing one used for the community analyses. It also would have been good if kits from other manufacturers had been included, as the Qiagen kits tested are among the most expensive but are not always found to perform best.

The reason why we use Blood and Tissue kit for pure cultures and Powersoil kit for the community analyses is because they are amongst the best kits in their associated sample types. Blood and Tissue kit is considered as one of the best for pure cultures, but not microbial communities. This kit has been optimized for pure cultures of gram positive and gram negative bacteria. However, it does not include protocol for the microbial communities. (see protocol:

<https://www.qiagen.com/us/resources/download.aspx?id=68f29296-5a9f-40fa-8b3d-1c148d0b3030&lang=en>) So it is not recommended for microbial community analyses, although I found that some people use the gram positive protocol for microbiome samples. Similarly, Powersoil is optimized for analyzing microbial communities, in particular, the environmental samples. And it is considered as one of the most widely used kits for extracting gDNA from microbial communities. We want to choose the most appropriate kit in each case as the standard for pure cultures or microbial communities analysis.

• In the discussion the authors discuss the same points over and over, repeating themselves many times, but do not devote any space to discussing the limitations of this method or any other alternatives.

The whole discussion has been revised. Briefly, since direct PCR method carries everything in the sample into the PCR reagent, it is critical to make sure all the PCR inhibitors in the sample have been denatured and will not inhibit PCR. This is the major limitation. In addition, the recent protocol has low efficiency on bacteria with thick peptidoglycan, for example, stationary phase gram positive strains in rich medium as we mentioned in our manuscript. We are working on compatible enzyme treatment to solve this problem. According to the alternative, to our best knowledge, the only reported one is the Extract-N-Amp kit (Ref. 16 in our manuscript), which also shows similar efficiency to Powersoil but much more expensive than our method. It has been discussed in L. 236-256.

• The authors should provide the data from sequencing runs (deposited in e.g. NCBI SRA) as well as data and scripts used for analysis.

We have updated the data availability (L. 524) The composition of the microbial community standard samples and water samples are provided in Table S2 and Figure S2. The raw sequencing data is made available at Figshare (<https://figshare.com/s/d31777db4915ee7563da>). The code is available on Github (https://github.com/fasong/Direct_PCR_4_bacterial_community_analysis.git)

Minor comments

• Line 16: amount -> number

It has been corrected.

• Line 18: "biased on the types of species" is incomplete

It has been corrected.

• Line 49: what outcome?

We have revised it as “It is often the goal to use these data to find associations between community composition and biological function.”

• Throughout: Please use 16S (capital 'S') consistently

We have corrected them.

• Line 68-69: I agree that this information can be difficult to find, but it is not impossible and all kits follow a relatively similar formulation so I disagree with this statement.

Yes, we agree it is hard but not impossible. To avoid the misunderstanding, we decided to remove this statement.

• Lines 78-88: Please include the Blood & Tissue kit in this description.

We appreciate this suggestion. The reason we did not include Blood and Tissue kit is because we want to focus on microbial community analysis. And Powersoil is recommended for microbial community analysis. Blood and Tissue in our study is used for evaluating the efficiency of the direct PCR methods on different types of species. Since we do use Blood and Tissue kit in our study, we have included the Blood & Tissue kit in this description as you suggested (L. 81).

• Line 325: in the -80 -> at -80

It has been corrected.

• Line 332: standards was -> standard was or standards were

It has been corrected.

• Line 340: Methods elsewhere are specific about volumes used, but they are missing in this paragraph.

This session of direct PCR focuses on the description of the direct PCR methods but not specific experiments. The volumes used in each experiment are stated in their own session. For example, the quantitative PCR session mentions the volumes used in qPCR of lab enrichments. And the 16S amplicon PCR session covers the volumes used in the direct PCR for analyzing the microbial community standard and water samples.

• Line 349: 300 g -> 300 x g?

It has been corrected.

• Line 352: Genome DNA -> Genomic DNA

We have corrected it thoroughly.

• Line 361-366: I found this paragraph very hard to follow. Please restructure/rewrite. It would also be useful if the qPCR and MiSeq steps (as well as use of different pure cultures/communities) were added to Figure 1 so that it is easier to see how these both fit in.

We have revised the related paragraph (L. 366). In addition, because qPCR assay is used for evaluating the efficiency of direct PCR on various types of species (gram positive and negative), but not for the microbial

community analysis, we did not include it in Figure 1. Figure 1 is the scheme of the pipeline for microbial community analysis. So we feel it would be better to keep this Figure 1. The procedure of qPCR of pure culture follows the same pipeline shown in Figure 1 (just change PCR to qPCR).

• Line 394: Are the read lengths given here correct? Why did they deviate from the standard 2x300?

The read lengths here are correct. We used the previous reported protocol in our lab but not the Earth Microbiome Project protocol. The method we used is adopted in this reference - Justice, N. B., Sczesnak, A., Hazen, T. C. & Arkin, A. Environmental selection, dispersal, and organism interactions shape community assembly in high-throughput enrichment culturing. *Appl. Environ. Microbiol.* 83, e01253–17 (2017).

• Line 397: Is this referring to mismatches in the primer sequences? Please be specific.

No, it is not because of the mismatches in the primer sequences during amplification. It is because of the mismatches of the overlapping region of read 1 and read 2 during the assembly of forward and reverse reads. And the mismatches are caused by the low quality reads in the end of the read 2. We don't know the exact reason. But we do find the read quality drops significantly during long reads, in particular in the 600 bp sequencing protocol in Miseq. In addition, another possible reason is the customized read 2 in this protocol is not optimized. We have revised it accordingly to avoid miscommunication.

• Line 108-113: Why are these other primers not also included in the methods? There are also no details of what was used as a standard curve for the qPCR and therefore of how qPCR efficiency was checked and the values were normalised.

The primers in Figure S1(A) were used for screening the best primers for our qPCR assay. Luckily, any pair of these primers worked. So we just picked one for our evaluation. To evaluate the efficiency of our direct PCR method, we compared the Ct measured from the direct qPCR method with the Ct from normal qPCR after genomic DNA extracted by Blood and Tissue kit. We use the extracted genomic DNA from the same amount of cell cultures for the standard. So if the Ct are the same between direct PCR of the cell culture and the normalized qPCR of the extracted genomic DNA from the same amount of the cell culture, the direct PCR method have the similar efficiency on releasing gDNA out of the cell to Blood and Tissue kit. What we need is the relative value (higher or lower) but not the exact value, so we don't think the standard curve is necessary to answer our question – whether the direct PCR method has similar efficiency to Blood and Tissue kit or not.

• Figure 2: I initially found this very difficult to understand. I think it would drastically be improved by not showing the Blood & Tissue results 3 times, but rather grouping by sample so that the results of each different method can easily be seen alongside one another. It also would be useful to show the change in Ct values between Blood & Tissue and the direct PCR methods as a separate panel. Please also ensure that axis text labels are sufficiently separate from one another to be clear on any revised figure.

We really appreciate your suggestion, and totally agree that the figure would look better by grouping by methods. We have revised this figure 2. The changes of Ct values and *p* values (t-test) has been shown in Table S1.

• Lines 127-136: I found it difficult to assess whether I agree with the interpretation of the authors here, but I disagree with the statement that they performed similar to or better than the kit - the only case where this is true is for the gram negative with Method 3. The other methods are similar in some cases, but also worse in many.

We think there is a misunderstanding. In the paragraph L125-133, we talked about gram negative species, but not gram positive species. So the statement is based on gram negative strains as we said “All the three methods could

effectively interrupt the gram negative bacteria (*Escherichia coli*, *Pseudomonas putida*) in both exponential and stationary phases, similar to or better than the DNeasy Blood and Tissue kit.” We talked about gram positive species in the next paragraph.

• Line 131-132: Please specify that this is only for gram negative.

We stated it as “in this case” which means gram negative in our manuscript. And the whole paragraph is talking about gram negative strains. To avoid miscommunication, we have revised it as suggested.

• Paragraph starting on line 156: I am not sure why the authors chose to use Euclidean distance here, especially as they later use Bray-Curtis distance. I think that the Bray-Curtis distance is the more appropriate to use here, and they should use this instead.

We want to choose the most appropriate method for our analysis. Euclidean distance is the true distance between two samples, which is always the first choice to analyze the similarity/dissimilarity. Bray-Curtis distance is not the true distance since it does not satisfy the triangle inequality axiom. This is why some people suggest using the term “Bray-Curtis dissimilarity” but not “Bray-Curtis distance”. However, for the samples with many zero or low abundant species, Bray-Curtis is better than Euclidean because it can limit the influence of small numbers on the results. In our case, the real composition of our microbial community standard is known to be non-zero, so we choose the Euclidean distance. The composition of water samples contains many low abundant/zero abundant species, thus Bray-Curtis dissimilarity is chosen in that case. We believe our choice is appropriate. Although we chose to stay with the Euclidean distance, we also did Bray-Curtis dissimilarity on our microbial community standard results (As shown below, they represent the Figure 3 (B and C) in our manuscript using Bray-Curtis dissimilarity instead of Euclidean distance.), and the result is similar to the Euclidean distance.

• Line 182: I'm not really clear on why this fold bias has been used rather than just log2 fold change or something similar that is typically used to look at differences in abundance between samples.

We appreciate this comment. We believe fold bias is better, because it could reflect and emphasize the effects on low abundant species. In this paper, we want to see the bias on different species, not only the overall difference among different results which has been shown in the pCA plot.

• Lines 195-200: Please rephrase as this part was hard to understand what was meant.

This paragraph has been revised. Briefly, we evaluated the repeatability/reproducibility of the replicates with direct PCR or Powersoil using Euclidean distance as the indicator. This method is adopted by the ref 16 - Videvall, E., Strandh, M., Engelbrecht, A., Cloete, S. & Cornwallis, C. K. Direct PCR offers a fast and reliable alternative to conventional DNA isolation methods for gut microbiomes. *MSystems* 2, e00132–17 (2017).

• Figure 3: Why are there more powersoil replicates than direct PCR? I didn't think differences in the number of replicates were discussed in the methods. It would also be nice to add the mean differences (both for each species and overall) to panel D and please clarify on how panel E was calculated - is it all vs all, or is it average of all others against the replicate in question.

The reason why there are more powersoil replicates than direct PCR is because the first three replicates of powersoil seem like the figure below. One of the samples is much different from the other two (which is shown as the dot is lower than another two.). To figure out if it is the system deviation of the method or experiment error, we added another 3 replicates. And it seems like it is the method deviation. This is consistent with our results - the direct PCR is more reproducible than Powersoil.

We have listed the relative abundance data in Figure 3(D) in Table S2. The difference could be calculated based on the table S2. We find the figure 3(D) is enough to show the overall difference we stated.

The Figure 3(E) was calculated based on all vs all. For example, if there are 6 replicates, they are the distances of all the possible pairs of these 6 replicates.

• Figure 4: please plot ESVs in the same order between the two samples, this makes it quite difficult to look at.

Figure 4 has been updated.

- Colours in stacked bars for both Figures 3 and 4 are probably not colour blind friendly.

Figure colors have been updated.

- Line 217: There was no mention of normalisation/log transformation for the data shown in Fig. 3, so why is it done now? Consistency between the analyses would be good.

In Figure 3, we analyze the results of microbial community standard using Euclidean distance that is calculated based on relative abundances without transformation.

- It would be good to give results of Adonis tests on community compositions.

It has been updated in the manuscript (L. 225 and L. 411).

- Line 244-257: Please rephrase and reduce this paragraph. Much of it is repeated.

We have updated this paragraph accordingly. The simplicity of the steps and the cost of time are two different factors. They are related but not the same. We discussed both in this paragraph.

- Line 252: Why mention Extract-N-Amp in the results and in Table 2? It was not compared here. Table 2 should also have Blood & Tissue added.

The reason is Extract-N-Amp is the only previously reported direct PCR method used in microbial community analysis. We have mentioned in L. 237-238. Because Extract-N-Amp also claims similar efficiency compared to Powersoil, we want to compare the cost with our direct PCR method. This is why we mention Extract-N-Amp in table 2. Because blood & tissue is not recommended for microbial community analysis, we did not compare it with other methods we proposed for microbial community analysis.

- Line 264: Replace accuracy with efficiency

We have made corrections thoroughly.

- Line 278: difficult lyse -> difficult to lyse

It has been corrected.

- Line 279-280: I do not agree at all with this statement. This is not what is shown by their results.

We appreciate this comment. This statement is not from our results, but from the cited reference (ref 24 in the manuscript). We think it is reasonable. If two methods are the same efficiently, cost of time and labor would be the key factors for our decision.

- Lines 285-296: I have never seen an experiment set up in the way described here, where each combination of species in a system is grown and sequenced. While I agree here that the time of the DNA extractions for this suggested experiment would be a lot, I suspect that the real issue stopping an experiment like this is the sequencing cost (much more than DNA extraction).

Here are some examples where people use this method.

1. Venturelli, Ophelia S., et al. "Deciphering microbial interactions in synthetic human gut microbiome communities." *Molecular systems biology* 14.6 (2018): e8157.
2. Pacheco, Alan R., Melisa L. Osborne, and Daniel Segrè. "Non-additive microbial community responses to environmental complexity." *Nature communications* 12.1 (2021): 1-11.
3. Goldford, Joshua E., et al. "Emergent simplicity in microbial community assembly." *Science* 361.6401 (2018): 469-474.
4. Sanchez-Gorostiaga, Alicia, et al. "High-order interactions distort the functional landscape of microbial consortia." *PLoS biology* 17.12 (2019): e3000550.

Recently, it is difficult to do drop-off or add-on experiments by removing or adding one species in a microbial community to investigate the influence of each species on the assembly. One of the reasons is the cost of time and labor I mentioned in the manuscript.

We agree sequencing is costly. However, it has been cheaper and cheaper due to multiplexing. Depending on the sequencing depth, more than 400 samples could share the same lane in Miseq (the cost will be even lower by multiplexing samples in Novaseq.). It costs about \$1600 for extracting 384 gDNA by Powersoil, and the sequencing cost by Miseq is actually cheaper (~\$1200, 300 bp v3 kit.). Our method will decrease the gDNA extraction cost of 384 samples down to \$2. (All the costs do not include labor.)

• Lines 290-293: while there are some studies suggesting absolute quantification of microbial abundances, these are contested within the microbiome community and if the authors want to make statements like this then the other side must also be discussed. Typically, sequencing is not accepted as absolute quantification. Even with the inclusion of other marker genes or qPCR, copy number can vary drastically between even closely related species.

We agree it is difficult to get extreme accurate absolute abundance. However, it does not influence our statement. We stated that combining our direct PCR method with the sequencing based cell number quantification will be a unique high throughput way to study microbial assembly. Sequencing based methods (such as 16s sequencing) are traditionally not used for quantifying cell number (or absolute abundance). But with the improvement of cell number quantification by sequencing based methods, we believe it will become a popular tool due to its super high throughput and the ability to differentiate species. We did not state that the recent sequencing based cell number quantification method is super accurate, but just want to highlight the potential of this method in the future.

• Lines 300-301: There is a large assumption here that all of the DNA in a cell will be available for PCR, and I am not convinced this is true with these methods.

I will rephrase this sentence by replacing “all the DNA would be PCRred without any loss during the DNA extraction” with “keeping all the gDNA in the PCR without any loss during the DNA extraction” in L. 269-270.

July 6, 2021

Dr. Fangchao Song
Lawrence Berkeley National Laboratory
Berkeley

Re: mSystems00224-21R1 (A simple, cost-effective and automation-friendly direct PCR approach for bacterial community analysis)

Dear Dr. Fangchao Song:

Thank you for submitting your manuscript to mSystems. We have completed our review and I am pleased to inform you that, in principle, we expect to accept it for publication in mSystems. However, acceptance will not be final until you have adequately addressed the reviewer comments.

Both reviewers agree that the manuscript is much improved, however, Reviewer 2 has several outstanding comments that I feel require a response by the authors before a final decision can be made.

Preparing Revision Guidelines

For complete guidelines on revision requirements for your article type, please see the journal Article Types requirement at <https://journals.asm.org/journal/mSystems/article-types>. **Submissions of a paper that does not conform to mSystems guidelines will delay acceptance of your manuscript.**

Sincerely,

Sarah Glaven

Editor, mSystems

Journals Department
Reviewer comments:

Reviewer #1 (Comments for the Author):

My comments have been adequately addressed.

Reviewer #2 (Comments for the Author):

I thank the authors for their responses, the amendments and clarifications made to the paper and for providing their sequencing data and code used for analyses. Many improvements have been made but I do still have some remaining concerns. There are also several instances where I think the choice of wording given by the authors leaves room for misinterpretation and could be clarified. I have detailed an issue below, but I think a careful read through for clarity/grammar in much of the paper - but particularly in the discussion - may alleviate some of my concerns regarding the presentation of some ideas. For instance, line 296-299 (and this paragraph) implies that direct PCR can be used for quantification of absolute abundances (which I and others still strongly disagree with, e.g. <https://elifesciences.org/articles/46923>, <https://www.ncbi.nlm.nih.gov/pmc/articles/PMC6586903/>, <https://microbiomejournal.biomedcentral.com/articles/10.1186/s40168-021-01059-0>). Even if we were able to get at the total DNA content of a sample through sequencing approaches alone, this would still not give us an accurate idea of the number of cells or even the number of genomes because it would still not account for e.g. 16S copy number variation, genome copy number variation. In the context that these statements are made - with a community where all members are known and presumably information on the genetic content of each member is available, some of their statements may stand, but this needs to be stated explicitly. Please revise this paragraph accordingly. I also think some qualifiers should be added to some statements, for example, add "for the sample types we tested" to L234-235 "are comparable with the widely used DNA extraction kits". Also, in the abstract where it says "exhibits a comparable overall efficiency to the conventional...", please switch this for something like "exhibits a comparable efficiency in some circumstances...".

I have just one other comment that I apologise for not making in my first review - but I think it would be very interesting to know how the results already obtained would compare with the results from using a compositional framework for analysis (i.e.

<https://www.ncbi.nlm.nih.gov/pmc/articles/PMC6755255/>) - so with conversion of abundance values to centered log ratio (CLR) or robust centered log ratio (rCLR - for dealing with sparsity). I think that this could be particularly useful for the fold bias/fold change of specific bacterial species, although use of Aitchison distance (Euclidean on CLR transformed) or robust Aitchison distance would alleviate my concerns slightly, although the use of the "real" composition or abundance I do still think should be made clear in the text, as mentioned above.

Other minor concerns

- In several places the plural is lacking, e.g. "to powersoil kit" -> "to the powersoil kit" (several places), "bacteria phyla" -> "bacterial phyla" (L101), "bacterial interaction" -> "bacterial interactions" (abstract)
- In the abstract, please be specific that the Blood & Tissue was used with isolates and Powersoil with communities, i.e. "By comparing direct PCR methods with DNeasy blood and tissue kits and DNeasy powersoil kits" -> "By comparing direct PCR methods with DNeasy blood and tissue kits for isolates and DNeasy powersoil kits for communities"
- Unless I'm mistaken, I think that *L. fermentus* needs to be changed to *L. fermentum* throughout.
- Thank you for explaining the difference in the number of replicates for the methods to me, but I think this should also be added to the manuscript somewhere as this is likely to confuse others also.
- I am still confused about the 1x350 and 1x250 bp reads, as this is not the same as the previous study that you cite, which says 2x300.
- In the final section of the results the link between the methods described here and the other methods (Extract-N-Amp and the Videvall method) is still not clear. They are not sufficiently introduced at first mention - e.g. combining some introduction with the descriptions of the methods all at once to give something along the lines of "we also compared the cost of our method with a previously described direct PCR method, with or without an additional step. The protocol for the Extract-N-Amp Plant PCR kit includes extraction solution addition, heating treatment, and dilution solution addition steps (refs) and Videvall et al. modified this by also having an additional shaking/bead beating step using TissueLyzer (Qiagen) prior to/after (whichever is true) the Extract-N-Amp Plant PCR kit."
- I can agree that if the efficiency/accuracy is the same, then factors such as cost and time are most important, but this is not given in the text as a hypothetical (now line 270-271): "While the efficiency and precision are the same, the choice of methods are guided largely by the cost, time, and potential for automation. Thus the direct PCR method is a promising way of microbial community analysis" - this implies that they are the same. I think that changing this subtly to e.g. "If the efficiency and precision were the same, the choice of methods would be guided by the cost, time and potential for automation. Thus the direct PCR method is a promising method for microbial community analysis in certain situations".
- I think I didn't express myself clearly enough when I said that I had not seen an experiment set up in the way described there with multiple different combinations of species (originally L285-296). I didn't mean to infer that this had never been done, but rather that this is by far not the most common way to carry out an experiment and I am therefore not sure that a focus on this is the most relevant.

Responses to reviewer comments:

Reviewer #1 :

My comments have been adequately addressed.

Thanks. We really appreciate your review!

Reviewer #2 :

I thank the authors for their responses, the amendments and clarifications made to the paper and for providing their sequencing data and code used for analyses. Many improvements have been made but I do still have some remaining concerns. There are also several instances where I think the choice of wording given by the authors leaves room for misinterpretation and could be clarified. I have detailed an issue below, but I think a careful read through for clarity/grammar in much of the paper - but particularly in the discussion - may alleviate some of my concerns regarding the presentation of some ideas.

We really appreciate your suggestions. We have revised the paper. We hope it clarifies our idea.

For instance, line 296-299 (and this paragraph) implies that direct PCR can be used for quantification of absolute abundances (which I and others still strongly disagree with, e.g. <https://elifesciences.org/articles/46923>, <https://www.ncbi.nlm.nih.gov/pmc/articles/PMC6586903/>, <https://microbiomejournal.biomedcentral.com/articles/10.1186/s40168-021-01059-0>). Even if we were able to get at the total DNA content of a sample through sequencing approaches alone, this would still not give us an accurate idea of the number of cells or even the number of genomes because it would still not account for e.g. 16S copy number variation, genome copy number variation. In the context that these statements are made - with a community where all members are known and presumably information on the genetic content of each member is available, some of their statements may stand, but this needs to be stated explicitly. Please revise this paragraph accordingly.

We agree that the 16s sequencing based method could not accurately quantify the absolute cell numbers in the sample. However, there are some efforts which make the estimation better, such as measuring the total DNA content. One of them is to add a spike-in which is the small amount of bacteria/gDNA with known absolute abundance. By comparing the relative abundance of every strain with the relative abundance of the spike-in, we can estimate the absolute abundance. This is also mentioned in one of the papers you shared, <https://www.ncbi.nlm.nih.gov/pmc/articles/PMC6586903/>, - *"If the absolute abundance of one taxon and the relative abundance of all taxa is known, it is feasible to compute the absolute abundance of all taxa."* We admit that since there are still differences on 16s copy numbers and PCR efficiency between the strain and the spike-in, we still could not accurately quantify the absolute abundance, but many papers have accepted this level of bias, and call it absolute abundance, for example, <https://microbiomejournal.biomedcentral.com/articles/10.1186/s40168-018-0491-7>, <https://www.nature.com/articles/s41592-019-0467-y>, <https://onlinelibrary.wiley.com/doi/epdf/10.1002/mbo3.977>, <https://onlinelibrary.wiley.com/doi/pdf/10.1002/mbo3.1220>.

Also, due to the advance in sequencing based quantification and the decrease of the cost of sequencing, there will be the demand of using the sequencing based method to quantify the growth of strains in a community as an alternative way of the OD based or fluorescence-based growth curve measurement. One of the advantages of sequencing based quantification is the ability to target the growth of multiple strains simultaneously. To apply this to study the microbial community dynamics and assembly, it requires vast amounts of samples to cover various combinations,

various time points, and replicates. Our method is ideal to process huge amounts of samples in this scenario. So we say the direct PCR method combined with the technique of quantifying absolute abundance would allow us to study the dynamics of microbial communities.

In addition, when we use the sequencing based method to quantify the microbial growth, the bias of 16s copy number, PCR amplification would not be an issue because we only compare the “absolute abundance” (compared to spike-in) of the same strain. They have the same 16s copy number and PCR bias compared to the spike-in, so the bias could be cancelled out during the comparison of two time points.

To clarify our statement, we modified this paragraph L.291-L.307. Hopefully it is clear now.

I also think some qualifiers should be added to some statements, for example, add "for the sample types we tested" to L234-235 "are comparable with the widely used DNA extraction kits". Also, in the abstract where it says "exhibits a comparable overall efficiency to the conventional...", please switch this for something like "exhibits a comparable efficiency in some circumstances..."

We appreciate your suggestion, and made revision accordingly (L.21 and L.229). We thought the default is that every statement is based on everything which gets tested, because it is difficult to generalize any experimental results without further validation. But, yes, it is always better to make it more specific, in particular in this case you feel it may cause misinterpretation.

I have just one other comment that I apologise for not making in my first review - but I think it would be very interesting to know how the results already obtained would compare with the results from using a compositional framework for analysis (i.e. <https://www.ncbi.nlm.nih.gov/pmc/articles/PMC6755255/>) - so with conversion of abundance values to centered log ratio (CLR) or robust centered log ratio (rCLR - for dealing with sparsity). I think that this could be particularly useful for the fold bias/fold change of specific bacterial species, although use of Aitchison distance (Euclidean on CLR transformed) or robust Aitchison distance would alleviate my concerns slightly, although the use of the "real" composition or abundance I do still think should be made clear in the text, as mentioned above.

We appreciate this comment, and add a sentence to explain the real composition in L.335. According to the method of examining the difference among samples, we agree that there are many methods we can choose, and it would be very interesting to know the differences among different methods. However, it is out of the scope of this paper. And many bioinformatics papers compared different methods. Since Euclidean distance and Bray-Curtis distance are the most popular methods, we believe they are the best choice to demonstrate the distance in this paper.

To address your concern, we calculated the Aitchison distance (Euclidean on CLR transformed) for our standard community data, and the results are similar to the Euclidean results (Figure 3B). Please check the comparison below.

Aitchison Distance to Real Composition

Similarity

Other minor concerns

• In several places the plural is lacking, e.g. "to powersoil kit" -> "to the powersoil kit" (several places), "bacteria phyla" -> "bacterial phyla" (L101), "bacterial interaction" -> "bacterial interactions" (abstract)

We really appreciate the careful editing! And we have made changes thoroughly.

• In the abstract, please be specific that the Blood & Tissue was used with isolates and Powersoil with communities, i.e. "By comparing direct PCR methods with DNeasy blood and tissue kits and DNeasy powersoil kits" -> "By comparing direct PCR methods with DNeasy blood and tissue kits for isolates and DNeasy powersoil kits for communities"

We really appreciate the careful editing! And we have made changes accordingly (L.24).

• Unless I'm mistaken, I think that *L. fermentus* needs to be changed to *L. fermentum* throughout.

We appreciate your corrections! It is *L. fermentum*. We have corrected this typo (L.177, L.184, L.187).

• Thank you for explaining the difference in the number of replicates for the methods to me, but I think this should also be added to the manuscript somewhere as this is likely to confuse others also.

We added an explanation in L.193 .

• I am still confused about the 1x350 and 1x250 bp reads, as this is not the same as the previous study that you cite, which says 2x300.

Yes, this is the only modification from the previous study. We notified it in L.402. The reason we used the 1x350 Read 1 and 1x250 bp Read 2 is for obtaining high read quality overallly. The read 1 in this protocol has much lower

quality compared to read 1. And the quality of read 2 also decreases much faster than read 1. Therefore, to increase the overall read quality, we set up a longer read 1 and shorter read 2.

• In the final section of the results the link between the methods described here and the other methods (Extract-N-Amp and the Videvall method) is still not clear. They are not sufficiently introduced at first mention - e.g. combining some introduction with the descriptions of the methods all at once to give something along the lines of "we also compared the cost of our method with a previously described direct PCR method, with or without an additional step. The protocol for the Extract-N-Amp Plant PCR kit includes extraction solution addition, heating treatment, and dilution solution addition steps (refs) and Videvall et al. modified this by also having an additional shaking/bead beating step using TissueLyzer (Qiagen) prior to/after (whichever is true) the Extract-N-Amp Plant PCR kit."

We appreciate your suggestions, and have revised it accordingly in L.232-234.

• I can agree that if the efficiency/accuracy is the same, then factors such as cost and time are most important, but this is not given in the text as a hypothetical (now line 270-271): "While the efficiency and precision are the same, the choice of methods are guided largely by the cost, time, and potential for automation. Thus the direct PCR method is a promising way of microbial community analysis" - this implies that they are the same. I think that changing this subtly to e.g. "If the efficiency and precision were the same, the choice of methods would be guided by the cost, time and potential for automation. Thus the direct PCR method is a promising method for microbial community analysis in certain situations".

We appreciate your suggestions, and have revised it accordingly in L.267-268.

• I think I didn't express myself clearly enough when I said that I had not seen an experiment set up in the way described there with multiple different combinations of species (originally L285-296). I didn't mean to infer that this had never been done, but rather that this is by far not the most common way to carry out an experiment and I am therefore not sure that a focus on this is the most relevant.

Yes. It is not a common way people are doing. But this kind of experiment would be very useful to answer questions on microbial assembly. For example, how would a community change when a new species comes? Or how do the microbial interactions change during the presence of other microbes? We think the reason it is not the most common way is because of the huge cost. And the decrease of sequencing cost and the direct PCR method in this paper could make it affordable. We revised the paragraph to explain this (L.291-307).

August 30, 2021

Dr. Fangchao Song
Lawrence Berkeley National Laboratory
Berkeley

Re: mSystems00224-21R2 (A simple, cost-effective and automation-friendly direct PCR approach for bacterial community analysis)

Dear Dr. Fangchao Song:

Your manuscript has been accepted, and I am forwarding it to the ASM Journals Department for publication. For your reference, ASM Journals' address is given below. Before it can be scheduled for publication, your manuscript will be checked by the mSystems senior production editor, Ellie Ghatineh, to make sure that all elements meet the technical requirements for publication. She will contact you if anything needs to be revised before copyediting and production can begin. Otherwise, you will be notified when your proofs are ready to be viewed.

As an open-access publication, mSystems receives no financial support from paid subscriptions and depends on authors' prompt payment of publication fees as soon as their articles are accepted. =

Publication Fees:

We recognize that the video files can become quite large, and so to avoid quality loss ASM

suggests sending the video file via <https://www.wetransfer.com/>. When you have a final version of the video and the still ready to share, please send it to Ellie Ghatineh at eghatineh@asmusa.org.

Sincerely,

Sarah Glaven
Editor, mSystems

Journals Department
Figure S1: Accept
Supplemental Material: Accept
Figure S3: Accept
Supplemental Material: Accept
Figure S2: Accept
Figure S4: Accept
Supplemental Material: Accept